# Selective amide bond formation in redox-active coacervate protocells

Jiahua Wang[1,2], Manzar Abbas [1], Junyou Wang [3] ✉ & Evan Spruijt [1] ✉

Coacervate droplets are promising protocell models because they sequester a wide range of guest molecules and may catalyze their conversion. However, it remains unclear how life's building blocks, including peptides, could be synthesized from primitive precursor molecules inside such protocells. Here, we develop a redox-active protocell model formed by phase separation of prebiotically relevant ferricyanide ($Fe(CN)_6^{3-}$) molecules and cationic peptides. Their assembly into coacervates can be regulated by redox chemistry and the coacervates act as oxidizing hubs for sequestered metabolites, like NAD(P)H and gluthathione. Interestingly, the oxidizing potential of $Fe(CN)_6^{3-}$ inside coacervates can be harnessed to drive the formation of new amide bonds between prebiotically relevant amino acids and α-amidothioacids. Aminoacylation is enhanced in $Fe(CN)_6^{3-}$/peptide coacervates and selective for amino acids that interact less strongly with the coacervates. We finally use $Fe(CN)_6^{3-}$-containing coacervates to spatially control assembly of fibrous networks inside and at the surface of coacervate protocells. These results provide an important step towards the prebiotically relevant integration of redox chemistry in primitive cell-like compartments.

Amide bond formation is an essential chemical reaction in all forms of life that is catalyzed by highly evolved biomolecular machinery. However, before ribosomes and specialized enzymes became capable of protein synthesis[1,2], alternative, simple prebiotic routes to create peptide bonds in a spatiotemporally controlled way likely existed. Protocellular compartments provide a promising platform to localize chemical reactions relevant to life[3–5]. What the nature of such protocellular compartments capable of peptide synthesis could be, remains unknown. As plausible precursors to peptides, α-aminothioacids (AA-SH)[6] and acetylated α-amidothioacids (Ac-AA-SH)[7] have recently received attention. α-Aminothioacids (AA-SH) have excellent solubility and stability in aqueous solution and can undergo oligomerization in the presence of bicarbonate under alkaline conditions, but their prebiotic synthesis presents difficulties and their ligation at neutral pH is inefficient, as discussed recently[7]. α-Amidothioacids (Ac-AA-SH) could be synthesized under prebiotically relevant conditions in aqueous solution and ligated to α-aminonitriles with high yields. Thioacids are

generally considered interesting alternatives to biological thioesters for prebiotic peptide ligation, as their regioselective ligation into α-peptides can be catalyzed with the aid of an oxidizing agent, such as iron(III) chloride or ferricyanide[7–10]. However, relatively high concentrations of reactants or catalysts are often used for these oxidative peptide ligations, which may not have been easy to reach, and the ligation reaction is not localized in a cell-like compartment. Recent work has demonstrated that some micro-compartments and protocell systems could compartmentalize (bio)chemical reactions, and act as catalytic microreactors or reaction localization centers with potential for prebiotic peptide ligation[11–16].

In particular, membraneless compartments based on complex coacervates have been considered as versatile protocell models in the origins of life research[5,16–19]. Coacervate droplets are formed spontaneously by liquid-liquid phase separation, resulting in a polymer- or peptide-rich, cell-sized dense coacervate phase, and a coexisting dilute phase. These membraneless droplets can readily take up and

[1]Institute for Molecules and Materials, Radboud University, Heyendaalseweg 135, 6525 AJ Nijmegen, the Netherlands. [2]Department of Radiology, Shanghai Jiao Tong University School of Medicine Affiliated Sixth People's Hospital, Shanghai 200233, China. [3]State Key Laboratory of Chemical Engineering, East China University of Science and Technology, Shanghai 200237, China. ✉e-mail: junyouwang@ecust.edu.cn; e.spruijt@science.ru.nl

concentrate molecules from their surroundings, called guests or clients, due to charge complementarity or hydrophobicity[16,20,21]. The ability to concentrate and exchange guests make the coacervate droplets capable of supporting biochemical reactions, and sometimes enhancing the activity of catalysts, such as ribozymes[12,13], or enzymes in cascade reactions[15,22], thereby increasing the overall reaction rates. Recent work has demonstrated that coacervate droplets can be active[19,23], and that their formation and dissolution can be regulated by environmental changes, such as pH[24,25], temperature[26,27], light[28], enzymatic reactions[23,29–31], or fuel-driven chemical reaction cycles[19].

To date, research on coacervates as reaction localization centers has mostly focused on enzymatic reactions, or reactions involving complex RNAzymes (ribozymes). However, the fundamental biochemical reactions that lead to the formation of peptides, which themselves are often the building blocks of coacervates, have not been studied. For coacervates to be a plausible protocell model, they must be able to support a prebiotically relevant ligation reaction of peptides, and ultimately, link this reaction to the construction of new coacervates. Here, we show that $Fe(CN)_6^{3-}$-containing coacervates can potentially fulfill this role. These coacervates can be used to oxidize not only a variety of common metabolites, but also alpha-amidothioacids, which can subsequently be aminoacylated yielding a product with a new amide bond.

Coacervates with $Fe(CN)_6^{4-}$, the reduced form of $Fe(CN)_6^{3-}$, as multivalent anion have already been reported by Bungenberg-de Jong and Kruyt in 1929[32]. They used gelatin at low pH as a long polycation and observed small punctated droplets upon mixing with $Fe(CN)_6^{4-}$. However, the redox potential of the iron centers inside these coacervates remains unclear. More recently, it was shown that coacervates can be formed from much smaller cationic oligopeptides, such as oligolysine and oligoarginine with as little as five amino acids, complexed with either small tri- or tetravalent anions, like ADP and ATP[18]. We therefore hypothesized that it should be possible to produce coacervates of the redox couple $Fe(CN)_6^{3-}$ and $Fe(CN)_6^{4-}$ with a cationic polypeptide in such a way that $Fe(CN)_6^{4-}$, the reduced state with a 4− charge, forms stable droplets, but $Fe(CN)_6^{3-}$, the oxidized state with a 3− charge, does not. This would enable the selective compartmentalization of guest molecules only under certain redox potential in the environment. In addition, from an origins of life perspective, the environment on early Earth was likely depleted of oxygen, a strong oxidizing agent that is widely used in living systems, for a considerable period. Alternative oxidizing agents, such as $Fe(CN)_6^{3-}$, could have has been essential in the prebiotic activation of building blocks for peptide ligation[7,9,33–36], and oxidation of metabolites[37]. By condensing the $Fe(CN)_6^{3-}$ with an oppositely charged peptide at low ionic strength, it may be possible to localize these reactions in a droplet compartment and enrich the coacervate with the products of $Fe(CN)_6^{3-}$-mediated oxidation reactions.

Here, we show that both $Fe(CN)_6^{3-}$ and $Fe(CN)_6^{4-}$ ions can be condensed into coacervate droplets with short cationic polypeptides depending on the ionic strength. These droplets are responsive and can be regulated by redox chemistry. They act as prebiotic oxidizing hubs for metabolites and common electron donors, such as NADH, NADPH and GSH, thus taking the role of oxygen as terminal electron acceptor. Moreover, we show that the oxidizing potential of $Fe(CN)_6^{3-}$-based coacervates can be harnessed to drive the formation of peptide bonds between amino acids and alpha-amidothioacids which are considered as potential prebiotic precursors of amino acids. We demonstrate that thioacid ligation is enhanced in $Fe(CN)_6^{3-}$/peptide coacervate dispersions compared to the surrounding dilute phase due to the high local $Fe(CN)_6^{3-}$ concentration. The coacervate environment imposes a selection pressure that results in a strong preferential incorporation of certain amino acids from mixtures of amino acids. The preference for incorporation can be rationalized by a combination of the amino acid partition coefficients (local concentration), intrinsic

reactivities, and strength of the interactions with the coacervate matrix (activity coefficient). This combination can give rise to the situation that, despite equal local concentrations of two amino acids, one shows a more than tenfold enhanced incorporation in the final product. Besides oxidation of aminothioacids, $Fe(CN)_6^{3-}$-based coacervates can facilitate the localized oxidation or aromatic thiols that have a potential to self-assembly into stacked filaments, such as benzoyl cysteine. In this case, the oxidation reaction can be visualized by the appearance of stacked filaments, which further bundle into rigid fibrils that resemble a cytoskeletal network inside and around the coacervate droplets, demonstrating the $Fe(CN)_6^{3-}$-based coacervates can spatially control the oxidation and self-assembly of thiols and metabolites. In short, our results show that prebiotically relevant $Fe(CN)_6^{3-}$-based coacervate protocells are versatile oxidizing hubs that exist in aqueous solution, in which metabolites can be converted, peptides synthesized and fibrous networks assembled.

## Results and discussion

### Coacervation of $Fe(CN)_6^{3-}$/ $Fe(CN)_6^{4-}$ and polypeptides

Small, multivalent ions can be condensed into liquid coacervate droplets by complexation with an oppositely charged peptide or polymer[5]. We sought to use this principle to create coacervate protocells that can concentrate a prebiotically relevant redox catalyst to enable localized peptide synthesis. $Fe(CN)_6^{3-}$ is a trivalent anion and its reduced form, the tetravalent $Fe(CN)_6^{4-}$, has been shown to be able to form coacervates with gelatin at low pH[32]. We used cationic peptides to induce phase separation of $Fe(CN)_6^{3-}$ and $Fe(CN)_6^{4-}$. Spherical complex (heterotypic) coacervate droplets were readily formed as a turbid dispersion via spontaneous liquid-liquid phase separation associated with the charge neutralization of a range of peptides (($Lys)_{10}$, $(Lys)_{20}$, $(Lys(Me)_3)_{20}$, $(Lys)_{30}$, $(Lys(Me)_3)_{30}$, poly-L-lysine (pLys) and $(Arg)_{10}$) in the presence of $Fe(CN)_6^{4-}$ or $Fe(CN)_6^{3-}$ (Fig. 1a, Supplementary Fig. 4). $(Arg)_{10}$ formed coacervates with both $Fe(CN)_6^{4-}$ and $Fe(CN)_6^{3-}$, while $(Lys)_{10}/(Lys)_{20}$ only formed coacervates with tetravalent $Fe(CN)_6^{4-}$ in NaCl-free solution. The difference between $(Lys)_{10}$ and $(Arg)_{10}$ can be explained by the higher $pK_a$ of the basic residue in arginine compared to lysine, which may generate more stable ionic interactions at a given chain length[38]. Previous studies also showed that $(Arg)_{10}$ could form coacervates with both trivalent adenosine diphosphate (ADP) and tetravalent adenosine triphosphate (ATP) at a certain salt concentration, while $(Lys)_{10}$ could only form coacervates with the more strongly charged ATP at the same salt concentration[18,39]. Increasing the length and thereby the overall charge of the polylysine could rescue the coacervates with ADP[18]. Here, increasing the length of polylysine to $(Lys)_{30}$ resulted in the formation of coacervates with trivalent $Fe(CN)_6^{3-}$ (Supplementary Fig. 4), in agreement with previous studies.

The obtained coacervates with a diameter of about 2–3 μm were observed in a PLL-g-PEG functionalized microchamber (Fig. 1b, c). These microdroplets exhibited liquid-like properties: they wetted the bottom glass surface and sedimenting coacervates coalesced with coacervates already present at the bottom of the microchambers to form bigger droplets, and coacervate droplets showed rapid fluorescence recovery after photobleaching (FRAP, Supplementary Fig. 3). The droplets sequestered negatively charged client solutes, including pyranine, NAD(P)H, and ribonucleic acids (Fig. 1d–f, Supplementary Fig. 6).

Determination of the partitioning of $Fe(CN)_6^{3-}$ and $Fe(CN)_6^{4-}$ by Uv-vis spectroscopy (see Methods) showed that both these multivalent scaffolding anions were highly concentrated in the coacervate droplets: the internal concentration in coacervate droplets was 100 times higher than the surrounding dilute aqueous phase. For example, for samples prepared from 2 mM $Fe(CN)_6^{3-}$ and pLys (5 mM lysine monomers) solutions, we found an internal $Fe(CN)_6^{3-}$ concentration of ~30 mM, compared to ~0.3 mM in the surrounding aqueous phase (Supplementary Fig. 7). It is worth noting that the coacervate is in

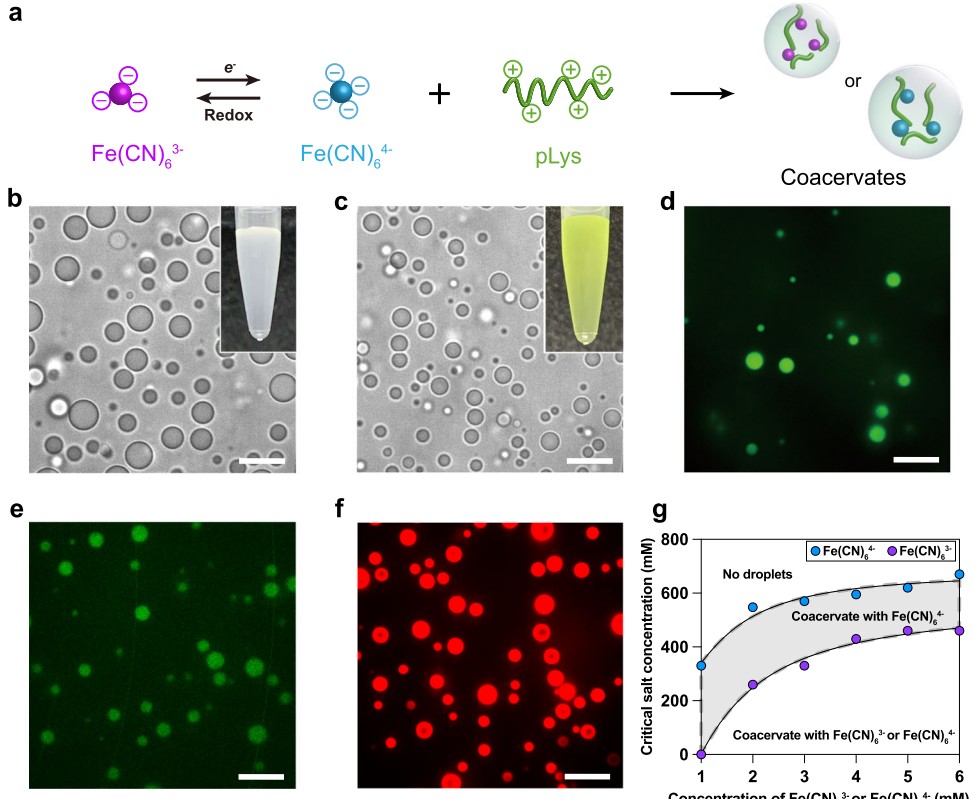

**Fig. 1 | Coacervation of Fe(CN)$_6$$^{3-}$/ Fe(CN)$_6$$^{4-}$ and polypeptides. a** Schematic illustration of associative liquid-liquid phase separation of Fe(CN)$_6$$^{3-}$/Fe(CN)$_6$$^{4-}$ and pLys to produce coacervate droplets. **b** Optical microscope images of Fe(CN)$_6$$^{4-}$/pLys droplets prepared at 1 mM Fe(CN)$_6$$^{4-}$ and 5 mM pLys (monomer basis), and (**c**) at 2 mM Fe(CN)$_6$$^{3-}$ and 5 mM pLys (monomer basis). Insets show photographs of the corresponding turbid suspensions. Scale bars, 10 μm. Fluorescence microscopy images of Fe(CN)$_6$$^{4-}$/pLys droplets with various client molecules: (**d**) pyranine, (**e**)

NADPH, (**f**) poly-rU$_{15}$. Scale bars correspond to 10 μm. **g** Critical salt concentration of Fe(CN)$_6$$^{3-}$ and Fe(CN)$_6$$^{4-}$ coacervates with a fixed concentration of 5 mM pLys, determined from turbidity titrations. Shaded regions indicate conditions under which coacervate microdroplets were observable. Solid lines are guides to the eye. The experiments in **b**–**f** were repeated three times with similar results. Each point in **g** represents an independently prepared sample.

thermodynamic equilibrium with the supernatant, meaning that the concentrations in both phases are maintained indefinitely. Increasing the concentration of pLys and ferricyanide (while keeping their ratio constant) would only increase the total volume of the coacervate phase and reduce the volume of supernatant (Gibbs' phase rule). Surprisingly, the concentration ratio for Fe(CN)$_6$$^{3-}$ and Fe(CN)$_6$$^{4-}$ was the same, despite their different valency. We attribute this to the formation of a tighter complex between Fe(CN)$_6$$^{3-}$ and polycations, which results in a more hydrophobic coacervate[27], an effect that has previously been observed with Fe(CN)$_6$$^{3-}$ in polyelectrolyte brushes[40]. A similar 100x higher local concentration was previously found for a negatively charged bis-carboxylic acid ligand, which can exist in a tetravalent (4-) ring configuration, in coacervates with cationic polymers[41]. These initial observations suggest that the spontaneous assembly of peptides and Fe(CN)$_6$$^{3-}$ or Fe(CN)$_6$$^{4-}$ could be developed as potential prebiotic membraneless compartments.

Typically, coacervates formed by charge-charge interaction are sensitive to the ionic strength and they exhibit a critical salt concentration (CSC) above which phase separation does not take place. Since Fe(CN)$_6$$^{3-}$ and Fe(CN)$_6$$^{4-}$ have a different net charge, their CSC is likely different, even though the previously discussed hydration differences may decrease that effect. A difference in CSC would allow for selective compartmentalization controlled by redox chemistry. To determine the conditions under which the redox couple Fe(CN)$_6$$^{3-}$ and Fe(CN)$_6$$^{4-}$ could give rise to reversible coacervate formation and dissolution, we evaluated the salt resistance of Fe(CN)$_6$$^{4-}$ and Fe(CN)$_6$$^{3-}$-based coacervates. As a model peptide, we focused on pLys ($M_w$ = 15−30 kDa). Supplementary Fig. 7 shows turbidity-based titration

curves of pLys (5 mM monomer units), as a function of Fe(CN)$_6$$^{4-}$ or Fe(CN)$_6$$^{3-}$, and as a function of salt concentration. From plots of the turbidity we determined the critical salt concentration (CSC), the point at which coacervate droplets completely disappear (see Methods). Figure 1g shows the resulting phase diagram of both Fe(CN)$_6$$^{4-}$ and Fe(CN)$_6$$^{3-}$ coacervates. As expected, Fe(CN)$_6$$^{4-}$-based coacervates have a higher salt resistance, expressed by their CSC, compared to Fe(CN)$_6$$^{3-}$-based coacervates. Our results are in good agreement with previous studies with nucleotide/pLys-based coacervates, where the ATP/pLys droplets have a higher CSC than ADP/pLys droplets, in line with their valency[18,30].

## Redox chemistry of Fe(CN)$_6$$^{3-}$/ Fe(CN)$_6$$^{4-}$ in coacervates

We next exploited the redox activity of the droplets in the critical salt concentration window highlighted in Fig. 1g. To illustrate the feasibility of the redox cycling proposed in Fig. 2a to induce phase separation and droplet dissolution, we prepared mixtures of Fe(CN)$_6$$^{4-}$ with pLys and Fe(CN)$_6$$^{3-}$ with pLys with identical pH and concentrations of all other components within the highlighted region of Fig. 1g between the two binodal lines. For example, the original Fe(CN)$_6$$^{4-}$-containing mixtures (Fe(CN)$_6$$^{4-}$ 1 mM, pLys 5 mM (monomer basis, 15−30 kDa), NaCl-free, $V_t$ = 200 μL) and Fe(CN)$_6$$^{3-}$-containing mixtures (Fe(CN)$_6$$^{3-}$ 1 mM, pLys 5 mM (monomer basis, 15−30 kDa), NaCl-free, $V_t$ = 200 μL) are white turbid and yellowish transparent, respectively. When observed under the microscope, the Fe(CN)$_6$$^{4-}$-containing mixtures had clearly condensed into droplets, while the Fe(CN)$_6$$^{3-}$-containing mixtures remained a homogeneous solution. Upon oxidation of the Fe(CN)$_6$$^{4-}$-coacervates by 0.5 equivalent of S$_2$O$_8$$^{2-}$ (1 μL, 100 mM), the originally

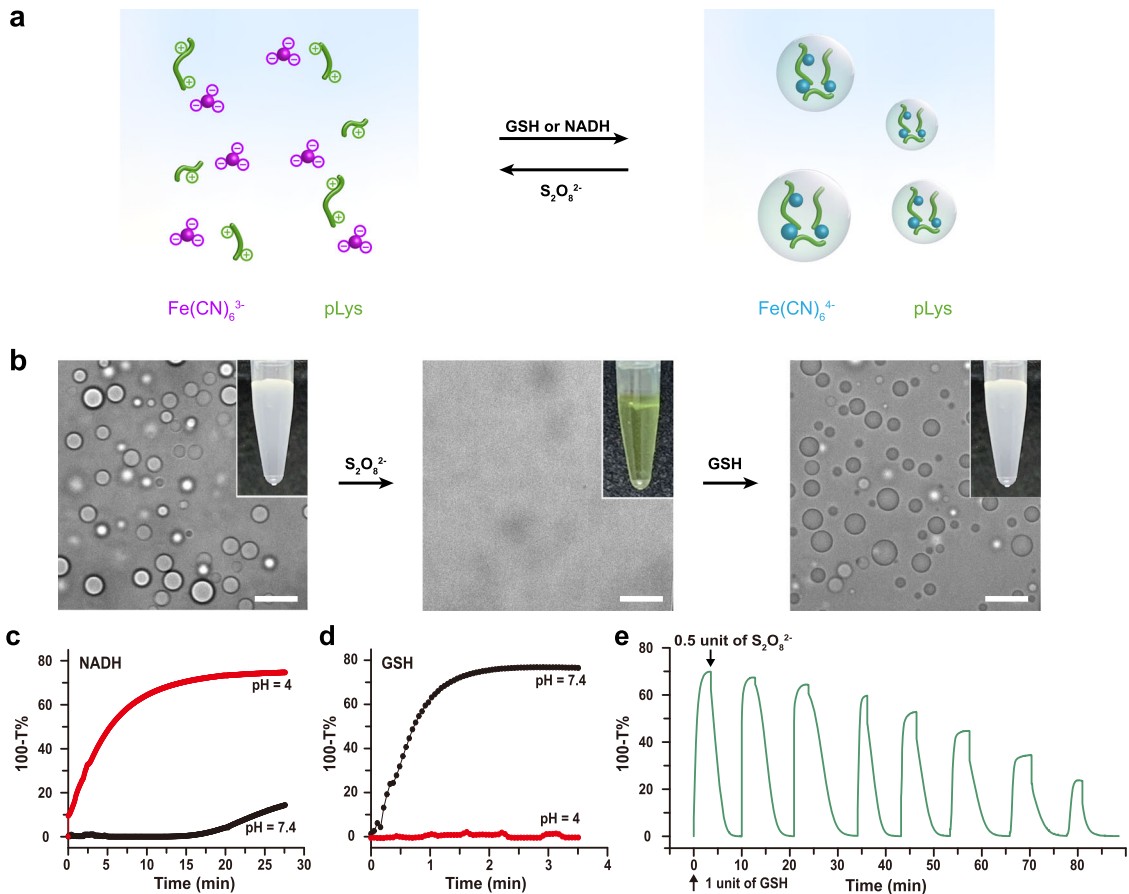

**Fig. 2 | Redox chemistry of Fe(CN)$_6^{3-}$/ Fe(CN)$_6^{4-}$ in coacervates. a** Schematic illustration of the redox reaction network underlying dynamic and reversible formation and dissolution of Fe(CN)$_6^{4-}$/pLys coacervate droplets. **b** Optical observation of droplet dissolution by 0.5 eq. of S$_2$O$_8^{2-}$ addition to Fe(CN)$_6^{4-}$/pLys coacervates dispersion and droplet formation by 1 eq. of GSH addition. Scale bars, 10 μm. Insets show photographs of the corresponding turbid suspensions, or the clear solution. **c** Formation of Fe(CN)$_6^{4-}$/pLys coacervate droplets induced by addition of 1 eq. of NADH at pH 4, but not at pH 7.4, which we attribute to the spontaneous saturation of NADH at low pH and concurrent shift of its redox potential[42,52,53]. **d** Same as **c**, formation of Fe(CN)$_6^{4-}$/pLys coacervate droplets induced by addition of 1 eq. of GSH, which proceeds at pH 7.4, but not at pH 4, because oxidation of GSH occurs via the deprotonated form. **e** Alternating additions of GSH and S$_2$O$_8^{2-}$ at pH 7.4 show that condensation and dissolution are both reversible and that the system can be switched multiple times between a compartmentalized droplet state and a single-phase homogeneous solution. The experiments in **b**–**e** were repeated three times with similar results.

white turbid solution turned yellowish transparent, and no droplets could be observed under the microscope (Fig. 2b). Conversely, after reduction of the Fe(CN)$_6^{3-}$-containing mixtures with 1 equivalent of glutathione (GSH) (2 μl, 100 mM) or 1 equivalent of NADH (2 μl, 100 mM) (Fig. 2b–d), the original, light-yellow transparent Fe(CN)$_6^{3-}$-containing mixtures became white turbid, and droplets were clearly visible under the microscope.

The oxidation reaction showed a clear pH dependence (Fig. 2c, d). We were able to use a stoichiometric amount of GSH to turn a homogeneous Fe(CN)$_6^{3-}$ solution into a dispersion of Fe(CN)$_6^{4-}$ droplets in the presence of pLys within 5 min in neutral conditions (pH 7.4, Fig. 2b, d), while no conversion occurred under acidic conditions (pH 4), because oxidation of GSH occurs via the deprotonated form. When using NADH as reducing agent, almost no conversion of the same homogeneous Fe(CN)$_6^{3-}$ solution with pLys could be observed in neutral conditions (pH 7.4, Fig. 2c), while complete conversion into Fe(CN)$_6^{4-}$ droplets was observed within 15 min under acidic conditions (pH 4), which we attribute to the spontaneous saturation to the spontaneous saturation of the 5-6 double bond in the pyridinic ring in NADH at low pH, and a concurrent shift in redox potential. In both cases, with GSH and NADH, a stoichiometric amount of S$_2$O$_8^{2-}$ completely dissolved a dispersion of Fe(CN)$_6^{4-}$ droplets, converting Fe(CN)$_6^{4-}$ back into Fe(CN)$_6^{3-}$ within ten minutes. Figure 2e illustrates the remarkable reversibility of this process: droplets could be

generated and dissolved up to eight times, and we were able to carry out identical transitions when starting from either Fe(CN)$_6^{4-}$ or Fe(CN)$_6^{3-}$. After eight cycles, the system loses its ability to condense into droplets, which is mainly caused by accumulation of the waste products from GSH or NADH and S$_2$O$_8^{2-}$. The level of redox control over droplet generation shown in Fig. 2 has not been achieved before and holds great promise for the development of dynamic protocell models.

For use of these Fe(CN)$_6^{3-}$ coacervates as potential oxidizing hubs in which oxidation reactions could be localized, it is important to know where the redox reactions utilized in Fig. 2 take place. We can take advantage of the fluorescence of common redox-active metabolites such as NADPH, in combination with the observed pH dependence, to monitor the conversion Fe(CN)$_6^{3-}$ and Fe(CN)$_6^{4-}$. We incubated Fe(CN)$_6^{3-}$/(Arg)$_{10}$ coacervates (8 mM Fe(CN)$_6^{3-}$/24 mM (Arg)$_{10}$ monomer units) with 2 mM NADPH at neutral pH and observed clear NADPH fluorescence inside the coacervates, indicating that NADPH is sequestered by the coacervates (Supplementary Fig. 11). At this pH, the reduction of Fe(CN)$_6^{3-}$ by NADPH is suppressed (Fig. 2c), which allows for equilibration and focusing of the microscope. We then decreased the outer pH by addition of a fixed amount of acid and monitored how the redox reaction progressed and led to the rapid disappearance of the fluorescence of NADPH within 2 min. (Supplementary Fig. 11). Interestingly, the fluorescence intensity first disappeared from the

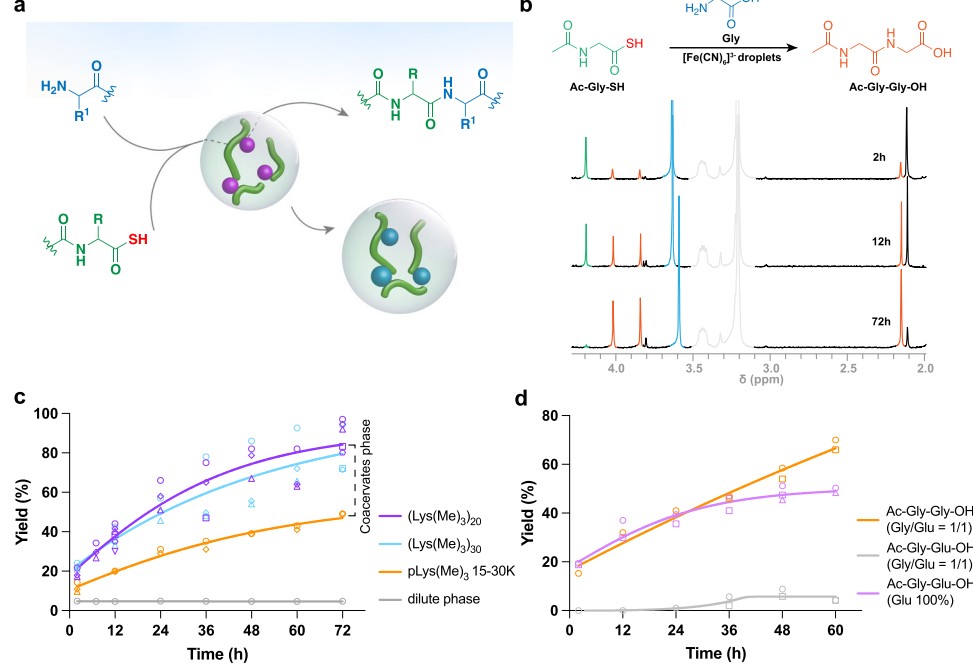

**Fig. 3 | Peptide bond formation in $Fe(CN)_6^{3-}$-based droplets. a** Schematic illustration of $Fe(CN)_6^{3-}$-based droplets as microreactors for peptide ligation in water. **b** $^1H$ NMR spectrum showing the peptide ligation reaction of $N$-acetyl-glycine thioacid Ac-Gly-SH (8 mM, green) and Gly-OH (3 eq., blue) with $Fe(CN)_6^{3-}$/pLys(Me)$_3$ (gray) coacervates (1 eq., pH 9, room temperature) to yield Ac-Gly-Gly-OH (orange). **c** Plot of % ligation products versus time for peptide ligation reaction in dilute phase and $Fe(CN)_6^{3-}$-based coacervate droplets. Each time point represents an independently prepared sample. Symbols: samples represented by circles, diamonds and triangles were prepared and analyzed together as a series; samples represented by squares, hexagons and inverted triangles were separately prepared and analyzed as a series. Different symbols correspond to yields determined from different peaks. **d** Plot of selective peptide bond formation within coacervate droplets. Each time point represents an independently prepared sample.

center of the coacervate droplets, and progressed radially outward on a typical timescale of 12 seconds (Supplementary Fig. 11), indicating that the oxidation of NADPH to non-fluorescent NADP$^+$ by $Fe(CN)_6^{3-}$ took place predominantly inside the droplets instead of in solution[19], in which case an exchange of NADPH/NADP$^+$ and reduction of fluorescence at the droplet/solution interface would be expected. Taken together, these data show our $Fe(CN)_6^{3-}$-based coacervates are redox-active compartments that can locally oxidize sequestered metabolites.

## Amide bond formation through α-amidothioacid oxidation in coacervate protocells

Having established the potential of $Fe(CN)_6^{3-}$-based coacervates as oxidizing hubs for redox-active guest molecules, we sought to use the oxidizing potential to synthesize peptides by catalyzing the formation of amide bonds. $Fe(CN)_6^{3-}$ has been described as a prebiotically abundant oxidizing agent[33–35], and has been used to directly activate α-aminothioacids and α-amidothioacids by oxidation to facilitate the formation of amide bonds[6,7,9], or to oxidize amino acid thiocarbamates[36]. Here, we investigated the oxidative aminoacylation of α-amidothioacids in the coacervates phase. The highly concentrated $Fe(CN)_6^{3-}$ coacervate droplets could be interesting model compartments for prebiotic amide bond formation, as they localize and concentrate the oxidizing agent and the presence of other coacervate components could give rise to a selective reaction of certain amino acids (Fig. 3a). As a proof of principle, we first prepared $Fe(CN)_6^{3-}$ coacervate droplets by direct mixing of aqueous solutions of $Fe(CN)_6^{3-}$ and trimethylated poly-L-lysine (pLys(Me)$_3$) at charge stoichiometry. The pLys(Me)$_3$ was chosen to avoid the ε-coupling of Lys-NH$_2$, although the p$K_a$ of the Lys ε-amino group is significantly higher than any α-amino group. To study the amide bond formation in coacervate droplets, we incubated $N$-acetyl-glycine thioacid (Ac-Gly-SH) (8 mM) with Gly (3 eq.) in $Fe(CN)_6^{3-}$ (1 eq.)/ pLys(Me)$_3$ coacervates dispersion

at pH 9 (±1). The consumption of Ac-Gly-SH and formation of the ligation product Ac-Gly-Gly-OH was monitored with $^1H$ NMR (Fig. 3b). As the reaction proceeds, the pH gradually decreases. We chose to maintain the pH by sequential addition of 0.2 μL aliquots of NaOD (1 M solution) while monitoring the pH (see Methods) instead of using a buffer, in order to avoid any destabilization of the $Fe(CN)_6^{3-}$-based coacervates with the shortest pLys by ions in the buffer. Control experiments with longer pLys-based coacervates in borate buffer (pH 9.2) showed the same ligation rates and yield (Supplementary Fig. 15).

Interestingly, we found a yield of Ac-Gly-Gly-OH of up to 80% after 3 days in the presence of $Fe(CN)_6^{3-}$/shorter peptides coacervates, while in dilute phase at most 5% of ligation product was observed. In controls with peptides but without $Fe(CN)_6^{3-}$, we found no ligation product (Supplementary Fig. 12), and also in the presence of $Fe(CN)_6^{4-}$/pLys coacervate droplets that are not capable of oxidizing Ac-Gly-SH, no ligation product was observed (Supplementary Fig. 13), demonstrating that neither the peptide nor the presence of coacervates alone facilitates the reaction. The significantly higher yield in the presence of $Fe(CN)_6^{3-}$-containing coacervates can be attributed to the high local $Fe(CN)_6^{3-}$ concentration inside coacervate droplets, and a locally enhanced concentration of the reactants (glycine has a partition coefficient of 9.4 in the coacervates, Table 1 and Supplementary Fig. 9), and implies that the ligation reaction takes place predominantly inside the coacervates. $Fe(CN)_6^{3-}$-based coacervate droplets can thus act as microreactors that enhance the rates of oxidative aminoacylation. We checked that coacervate droplets are still present at the end of the reaction after 60 h and that their round shape remained unchanged (Supplementary Fig. 5). We note that colloidal sulfur clusters are produced during the reaction, which appear as small dark spots with a typical size below the diffraction limit of the setup both inside and outside the coacervates (Supplementary Fig. 5). These clusters do not appear to influence the coacervate droplets.

**Table 1 | Mean volumes, $pK_a$, and coacervate partition coefficients (Supplementary Fig. 9) of amino acids used in this work**

| Amino acid (AA) | Volume (Å) | $pK_a$ | $K_p$ | Ligation yield to Ac-Gly-SH (%) |
|---|---|---|---|---|
| Gly | 63.8 | 9.58 | 9.4 | 72% |
| Ala | 90.1 | 9.71 | 8.8 | 32.5% |
| Glu | 140.8 | 9.58 | 30.2 | 49.5% |
| Phe | 193.5 | 9.09 | 23.1 | 54.7% |
| Asp | 117.1 | 9.66 | 36.7 | 47.9% |
| Asn | 127.5 | 8.76 | 21.0 | 40% |

Yields for the products of oxidative coupling of acetylated α-amido-thioacid Ac-Gly-SH (8 mM) with amino acid (24 mM) in $Fe(CN)_6^{3-}$ (8 mM)/(Lys(Me)$_3$)$_{20}$ (Lys monomer 24 mM) coacervates dispersion, pH 9, reaction time 60 h.

**Table 2 | Yields for the products of oxidative coupling of acetylated alpha-amido thioacid Ac-Gly-SH (8 mM) with a mixture of amino acids AA$_1$/AA$_2$ (ratio 1:1, 12 mM: 12 mM) in $Fe(CN)_6^{3-}$ (8 mM)/(Lys(Me)$_3$)$_{20}$ (Lys monomer 24 mM) coacervates dispersion, pH 9, unless stated otherwise**

| AA$_1$ | AA$_2$ | Yield ratio Ac-Gly-AA$_1$-OH (%): Ac-Gly-AA$_2$-OH (%) | |
|---|---|---|---|
| | | $t = 4\,h$ | $t = 60\,h$ |
| Gly | Glu | 15.3%: 0.0% (1:0) | 66%: 4.2% (16:1) |
| Gly | Ala | 25%: 4% (6:1) | 42.5%: 8.5% (5:1) |
| Gly | Phe | 26.6%: 8.0% (3.3:1) | 48.5%: 13.5% (4:1) |
| Glu | Ala | 19.5%: 11.5% (1.7:1) | 35%: 23% (1.5:1) |
| Glu | Phe | 23.3%: 13.7% (1.7:1) | 33%: 17% (1.4:1) |
| Gly | Asn | 26%: 5.3% (5:1) | 62.1%: 15.4% (4:1) |
| Gly | Asp | 25.4%: 6.6% (3.8:1) | 41.2%: 8.4% (4.9:1) |
| Asp | Asn | 24.1%: 17.9% (1.3:1) | 31.1%: 31.4% (1:1) |
| Asp | Ala | 20.0%: 13.6% (1.5:1) | 29.7%: 20.8% (1.4:1) |
| Asn | Ala | 14.6%: 12.2% (1.2:1) | 24.3%: 24.3% (1:1) |
| Asp | Phe | 18.2%: 3.7% (4.9:1) | 27.9%: 7.7% (3.6:1) |
| Asn | Phe | 22.8%: 4.6% (4.6:1) | 33.1%: 5.8% (5.7:1) |

We found that the ligation reaction was most enhanced in droplets formed from shorter peptides, (Lys(Me)$_3$)$_{20}$ ($M_w \approx 4$ kDa) and (Lys(Me)$_3$)$_{30}$ ($M_w \approx 6$ kDa), compared to longer pLys(Me)$_3$ ($M_w = 15$–30 kDa) (Fig. 3c). We attribute this to the lower multivalency of the polycation, which results in weaker complexation with the $Fe(CN)_6^{3-}$, making it more available for the reaction with the α-amidothioacids.

Powner and co-workers have shown that α-aminonitriles, such as Gly-CN, react more readily as nucleophiles with α-amidothioacids than amino acids, owing to their substantially lower $pK_a$ compared to amino acids[7]. Gly-CN could be ligated to Ac-Gly-SH over a broad pH range (pH 5–9), and Gly-CN was selectively incorporated over Gly and Gly-NH$_2$ in direct competition experiments. This superior reactivity made it possible to perform the ligation reaction even at reactant and $Fe(CN)_6^{3-}$ concentrations of several mM, although the reaction time increased to several hours. Therefore, we repeated the ligation reaction with Ac-Gly-SH and Gly-CN at different concentrations of reactants and $Fe(CN)_6^{3-}$. However, we did not observe any ligation product after 1 h at 8 mM of the substrate or less (Supplementary Fig. 37), while the ligation product had already formed with Gly in coacervates. These findings suggest that ligation reaction in our experiments is not primarily limited by the $pK_a$ of Gly, but by the availability of ferricyanide to oxidize the α-amidothioacids.

From an origins of life perspective, it is interesting to further explore this hypothesis of altered activity of certain reactants that gives rise to selectivity of the aminoacylation reaction by comparing the yield of ligation when more than one amino acid can react with an available thioacid inside the coacervate droplets. Instead of only glycine (Gly), competition reactions with stoichiometric (1:1) mixtures of two amino acids (glycine (Gly), alanine (Ala), phenylalanine (Phe), glutamic acid (Glu), aspartic acid (Asp) and asparagine (Asn)) were investigated. Many competition reactions demonstrated a significant selectivity for one of the ligation products at pH 9 (Table 2). For example, when we added both glycine (12 mM) and glutamic acid (12 mM) to $Fe(CN)_6^{3-}$-based droplets (8 mM $Fe(CN)_6^{3-}$) containing Ac-Gly-SH (8 mM), we observed a strong selectivity for peptide ligation with glycine. Glutamic acid alone yields the Ac-Gly-Glu-OH dipeptide in similar yields as glycine (Fig. 3d, Supplementary Fig. 16), but when glycine and glutamic acid were incubated together in a 1:1 ratio with Ac-Gly-SH in (Lys(Me)$_3$)$_{20}$/ $Fe(CN)_6^{3-}$ coacervates, more than 90% of the ligation products was Ac-Gly-Gly-OH (Fig. 3d, Supplementary Fig. 21). Glycine thus outcompeted glutamic acid very effectively in the ligation process. We reasoned that the activity of glycine inside coacervates is higher than glutamic acid. Glycine and glutamic acid have similar nucleophilicities and $pK_a$[42], the partition coefficient of Glu is higher than Gly (Supplementary Figs. 8 and 9, Table 1), resulting in a higher local concentration of Glu, and they reach similar yields when

reacted separately (Table 1). Nevertheless, Gly reacts more readily with Ac-Gly-SH, which suggests that the stronger interaction of Glu, which contains two carboxylic acid groups, with the oppositely charged peptides inside the coacervate lowers its activity and prevents it from reacting readily with Ac-Gly-SH. We note that the coacervate droplet presents a complex environment of which many physicochemical properties are not precisely known. Other factors, including steric effects (amino acids have different hydrodynamic radii), local pH and $pK_a$ differences, may also play a role in explaining this selectivity. The resulting behavior, that glycine reacts faster than glutamic acid inside the coacervate environment, can be interpreted as an example of kinetic pathway selection, caused by the local protocell environment, in which two species with the same intrinsic reactivity and concentration give rise to a selective formation of one product that is formed more rapidly due to a higher activity.

To further support the hypothesis that coacervates could give rise to selective incorporation of amino acids into dipeptides, we analyzed the product formation in other mixtures of two amino acids. Table 2 summarizes the results. We observed selective incorporation of glycine in mixtures of glycine and alanine (42.5% Gly/8.5% Ala), in mixtures of glycine and phenylalanine (45% Gly/14% Phe), in mixtures of glycine and aspartic acid (41.2% Gly/8.4% Asp), and in mixtures of glycine and asparagine (62.1% Gly/15.4% Asn) Table 2. Glutamic acid and aspartic acid are incorporated preferentially compared to phenylalanine (33% Glu/17% Phe and 27.9% Asp/7.7% Phe), indicating that the interactions between the aromatic phenyl group in Phe and the positive charges on the pLys chains outweigh the interactions between Glu or Asp and pLys. The preference for Gly over Asn, despite the fact that Asn has a significantly lower $pK_a$ compared to other amino acids (Table 1), suggests that Asn also interacts effectively with the coacervate components, possibly through hydrogen bond formation with the pLys backbone, which is further supported by its elevated partition coefficient (Table 1). It is interesting to note that polar, hydrogen-bonding amino acids are significantly enriched in many proteins that undergo coacervation, including FUS, hnRNPA1 and Ddx4[43], and that hydrogen bond formation was found to drive coacervation of disordered squid beak proteins[44], which suggests that hydrogen bonds can contribute substantially to the formation and stability of coacervates. We found no preference between aspartic acid and asparagine, also suggesting that hydrogen bond formation by Asn lowers its activity to a similar extent as the charge interaction of Asp. Finally, alanine was, perhaps

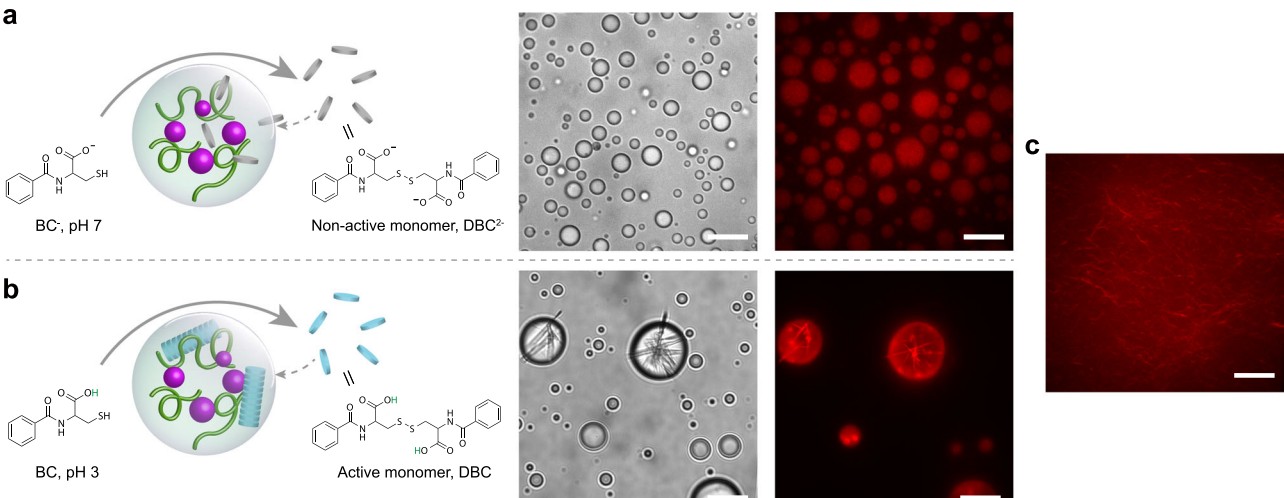

**Fig. 4 | Fiber self-assembly inside Fe(CN)$_6^{3-}$-based coacervates. a** Confocal micrographs of Fe(CN)$_6^{3-}$/(Arg)$_{10}$ protocells (with dye, Nile red), representative protocells after addition of BC$^-$. **b** Confocal micrographs of Fe(CN)$_6^{3-}$/(Arg)$_{10}$ protocells (with dye, Nile red) after addition of BC. **c** Confocal micrographs of fiber formation after adding Fe(CN)$_6^{3-}$ into BC solution. The experiments in **a**–**c** were independently repeated three times with similar results. Scale bars correspond to 10 μm.

surprisingly, not preferentially incorporated in any of the mixtures we tested (Table 2), while also showing a significantly lower yield when reacted separately (Table 1). Alanine is not expected to interact strongly with the coacervate, which is supported by its partition coefficient that is similar to Gly. Possibly, the lower reactivity is caused by the higher p$K_a$ of Ala, resulting in a more significant fraction of the Ala amino acids that is protonated under the experimental conditions.

## Fiber self-assembly inside Fe(CN)$_6^{3-}$-based coacervates

We further investigated if the redox activity of Fe(CN)$_6^{3-}$-based coacervates could lead to spatially controlled higher order assembly. Self-assembly of filaments inside or at the periphery of (proto)cellular compartments is key to many transport processes, locomotion and division[45,46]. Spatially controlled assembly of analogous model filaments in cell-like compartments is therefore an interesting goal in protocell and synthetic cell research[46–48]. We selected an amino acid derivative benzoyl cysteine as precursor for filaments (Fig. 4). *N,N*-dibenzoyl-L-cystine (DBC) is a well-known redox-active supramolecular gelator[49,50], which has been used to make filaments in aqueous solution at low pH. To create the precursor form, we reduced the water soluble DBC$^{2-}$ at pH 7 with 1 equivalent of dithiothreitol (DTT) to give non-active monomer *N*-benzoyl-L-cysteine (BC$^-$), which is highly soluble at both high and low pH and does not stack to form fibers.

We flushed a solution of BC$^-$ (with dye Nile red) in a microchamber containing Fe(CN)$_6^{3-}$/(Arg)$_{10}$ coacervates (final concentration: 10 mM BC$^-$, 4 mM Fe(CN)$_6^{3-}$, 12 mM (Arg)$_{10}$ (monomer basis)) at pH 7. Upon addition of the coacervates, we observed a gradual increase in the total fluorescent intensity inside coacervates, as newly formed non-active monomer 2- charged DBC$^{2-}$, which weakly binds to Nile red, partitioned in the coacervates (Fig. 4a). At this pH, no fibers are formed, and the non-active DBC$^{2-}$ monomers are distributed homogeneously inside the coacervates. We note that Fe(CN)$_6^{3-}$/peptide coacervates without DBC$^{2-}$ do not sequester Nile red (Supplementary Fig. 6c, d). Upon addition of BC to an aqueous dispersion of Fe(CN)$_6^{3-}$/(Arg)$_{10}$ coacervates at pH 3 (below the p$K_a$ of the carboxylate groups of DBC, p$K_a$ ∼ 3.6)[51], we observed oxidized, bright fluorescent DBC filaments assembled into shells around the coacervate droplets within 5 min, and several bundles of filaments present inside the droplets (Fig. 4b). The bundled filaments inside the coacervates are clearly visible in transmission, and are strongly stained by Nile red. These bundles can be seen to pierce the interface of the coacervate droplets. In control experiments with Fe(CN)$_6^{3-}$ but without polycations, we observed a fiber dispersion without clear bundling or formation of shells (Fig. 4c). The assembly of bundled filaments was reversible upon addition of β-mercaptoethanol to reduce the disulfide bond and also upon increasing the pH (Supplementary Figs. 35, 36), in agreement with previous reports[50].

A similar type of interfacial filament assembly has been observed for actin filaments and peptide-based pLys/pGlu coacervates before[40]. We also observed that different peptides lead to altered fiber assembly: in the case of Fe(CN)$_6^{3-}$/(Lys(Me)$_3$)$_{20}$ and Fe(CN)$_6^{3-}$/(Lys(Me)$_3$)$_{30}$ coacervates, the fibers preferentially localized inside and at the interface of the coacervates (Supplementary Fig. 34a, b). Interestingly, in the case of Fe(CN)$_6^{3-}$/(Lys)$_{30}$ coacervates, the fiber did not remain confined inside of the coacervates, but grew out into the surrounding environment, giving an aster-like shape (Supplementary Fig. 34c), possibly because the surface charge of the coacervates formed with non-methylated (Lys)$_{30}$ is different from that of coacervates formed with either (Lys(Me)$_3$)$_{30}$ or (Arg)$_{10}$, resulting in a different interaction with the growing fibers. In the case of the aster-like fiber growth, the resulting structures also appear more solid-like in general, possibly pointing at the presence of longer fibers than in the case of shell-like fiber growth. In short, these findings show that Fe(CN)$_6^{3-}$ coacervates can drive the formation of self-assembled filaments via oxidation of the filament precursors.

In summary, we developed a protocell model based on pre-biotically relevant Fe(CN)$_6^{3-}$ as redox-active species. Membrane-free droplet compartments were spontaneously assembled in the presence of short cationic peptides and the assembly of Fe(CN)$_6^{3-}$/ Fe(CN)$_6^{4-}$-peptide droplets can be regulated by redox chemistry and salt concentration. Fe(CN)$_6^{3-}$-peptide droplets can act as oxidizing hubs for metabolites, such as NAD(P)H and GSH, filament stacking element like benzoyl cysteine, and alpha-amidothioacids as potential prebiotic precursors of amino acids. We demonstrate that the oxidation of alpha-amidothioacids by Fe(CN)$_6^{3-}$ coacervates can be used to drive aminoacylation, resulting in the formation of new peptide bonds. The amino acid ligation is enhanced in coacervate dispersions compared to the surrounding dilute phase due to the local high Fe(CN)$_6^{3-}$ concentration. The coacervate environment imposes a selection pressure that results in preferential incorporation of amino acids that partition into the coacervates but display the least strong interactions with the coacervate matrix. Finally, this strategy can be used to drive the

spatially controlled assembly of fiber-like filaments by localized oxidation of building blocks inside coacervates. The filaments bundle into rigid fibrils that resemble a cytoskeletal network inside and around the coacervate droplets, which can be disassembled again by reduction or changing the pH. Our results show that prebiotically relevant $Fe(CN)_6{}^{3-}$-based coacervate protocells are versatile oxidizing hubs that exist in aqueous solution, in which metabolites can be sequestered and peptides synthetized. These results provide an important step towards prebiotically plausible integration of chemical processes in cellular compartments.

## Methods

### Materials
Poly-L-lysine hydrobromide (pLys, 15–30 kDa), potassium ferricyanide, potassium ferrocyanide, glutathione (GSH), DL-dithiothreitol (DTT), *N,N'*-dibenzoyl-L-cystine (DBC), tris(hydroxymethyl)aminomethane (Tris), pyranine, and sodium chloride were purchased from Sigma Aldrich and used without further purification. Short polycations $(Lys)_{10}$, $(Lys)_{20}$, $(Lys)_{30}$, $(Arg)_{10}$ were purchased form Alamanda Polymers. Nicotinamide adenine dinucleotide (NADH) and nicotinamide adenine dinucleotide phosphate (NADPH) were purchased from Roche. The fluorescently labeled oligonucleotides poly-$A_{15}$ (Cy5-$A_{15}$), Poly-r$U_{15}$ (r$U_{15}$-Cy3Sp) were purchased from Integrated DNA Technologies (IDT).

### Coacervate formation
Samples for turbidity measurements were prepared directly into 96-well plates, by adding, respectively, Milli-Q water, Tris buffer (pH 7.4, 50 mM), $(Lys)_n$/$(Arg)_{10}$, and ferricyanide or ferrocyanide to a total volume of 100 μL. Mixing was done by gentle pipetting (3×) before each measurement. Samples for the microscopy experiments were prepared in microcentrifuge tubes. After addition of the substrate, a 20 μL aliquot was immediately taken for imaging on a glass slide.

### Microscopy chambers preparation
The *Ibidi* μ-slides used for imaging were functionalized with PLL-g-PEG to minimize wetting and spreading of the coacervate droplets. Each slide was first activated by oxygen plasma treatment (Diener electronic, Femto). The PLL-g-PEG (0.01 mg/mL in 10 mM HEPES buffer, pH 7.4) solution was added into each well and incubated at room temperature for 1 h. After that, the glass slides were cleaned by rinsing (3 times with 10 mM HEPS, pH 7.4, 3 times with MQ). The slides were dried by using compressed air.

### Partitioning of dye molecules
For partitioning experiments, 20 μL aliquots of ferricyanide/pLys coacervates were added to PLL-g-PEG modified *Ibidi* μ-slides chambers. Small quantities of the stock solutions of the dye molecules were added to the multiphase coacervate droplets, mixed by gentle pipetting, and visualized by excitation at the specific wavelengths. Pyranine (0.2 μL, 0.1 mg/mL), NADPH (0.2 μL, 100 mM), NADPH (0.2 μL, 100 mM), poly-$A_{15}$ (0.2 μL, 1 mg/mL), Poly-r$U_{15}$ (0.2 μL, 1 mg/mL), and Nile red (0.2 μL, 1 mg/mL in ethanol).

### Turbidity measurements and critical salt concentration
Turbidity measurements were performed in triplicate using a Berthold Tristar (2) LB 942 microplate reader. The temperature was kept constant at $25 \pm 1\,°C$. The absorbance was measured at 520 nm, where none of the mixture components absorbed significantly. The absorbance of a well filled with the same volume of water was used as a blank. Samples were shaken for 5 s before every readout. The critical point was determined by extrapolating the first-order derivative at the inflection point to zero turbidity. Note that this critical salt concentration does not take into account ions from other sources than the added NaCl, and the actual critical ionic strength may be slightly higher.

### Ferricyanide and ferrocyanide partitioning
In a typical procedure, the coacervates dispersion was centrifuged in an Eppendorf tube at $3000 \times g$ until the dilute phase was transparent under microscope, the coacervate phase had sedimented to the bottom of the cell. The concentration of the ferricyanide/ferrocyanide in the dilute phase was quantitatively analyzed with Uv-Vis at wavelength of 320 nm and 420 nm (Supplementary Fig. 5). We measured the volume of the top (dilute) solution and calculated the ferricyanide/ferrocyanide concentrations of the top solution from the standard curve, the volume and the moles of ferricyanide /ferrocyanide in coacervate phase can be calculated from the total feed, then the ferricyanide and ferrocyanide concentrations inside the coacervate phase can be obtained.

### Amino acid partitioning
Stock solutions of amino acids (glycine, L-alanine, L-phenylalanine, L-aspartic acid, L-asparagine, L-glutamic acid and L-glutamine; 200 mM each), $(Lys)_{30}$ (200 mM), $K_3Fe(CN)_6$ (200 mM), NaCl (1.0 M) and 3-(trimethylsilyl)propionic-2,2,3,3-$d_4$ acid sodium salt (TMSP, 60 mM) were prepared in $D_2O$. Stock solutions of amino acids were brought to pH 10 (pH paper) using 1 M NaOD in $D_2O$. Phase separated samples were prepared by mixing 432 μL of $D_2O$, 72 μL of amino acid stock, 72 μL of $(Lys)_{30}$ stock and 24 μL of $K_3Fe(CN)_6$ stock. pH was adjusted to 9 (pH paper) using 1.0 M NaOD in $D_2O$. Samples were vortexed, transferred to PCV cell counting tubes (TPP Techno Plastic Products AG, Switzerland) and centrifuged at $14,000 \times g$ at 20 °C for 30 min. The volume of the coacervate phase was read from the scale on the cell counting tube. 550 μL of the supernatant was collected, mixed with 20 μL of TMSP stock and analyzed by NMR (Bruker Avance III 500 MHz). The supernatant remaining in the cell counting tube was discarded and the coacervate phase was dissolved by adding 600 μL of NaCl stock followed by aspirating with a syringe equipped with a needle several times. 550 μL of the re-dissolved coacervate phase was collected, mixed with 20 μL of TMSP stock and analyzed by NMR. Concentrations of amino acids were determined from $^1H$ NMR, using TMSP as a standard.

### Confocal fluorescence microscopy
Optical and fluorescence microscopy images were recorded on an Olympus UIS2 microscope, equipped with a motorized stage (Prior, Optiscan II). Fluorescent images were recorded with an EMCCD camera (Andor, iXon), using illumination from a mercury lamp, an excitation filter of 482/18 nm (Semrock BrightLine) and an emission filter of 525/45 nm (Semrock BrightLine). Samples were loaded into the wells of PLL-g-PEG-functionalized *Ibidi* μ-slides and closed with a lid (microscopy chambers).

### Fluorescence recovery after photobleaching
Fluorescence recovery after photobleaching (FRAP) experiments were conducted on an Olympus IX81 spinning disk confocal microscope, equipped with an Andor FRAPPA photobleach module and Yokogawa CSU-X1 spinning disk. Andor 400 series solid state lasers were used to bleach and image the samples. Images were recorded with a 100× oil immersion objective (NA 1.5) and an Andor iXon3 EM CCD camera. For FRAP measurements, we prepared samples as described above using 33% Alexa-488-labeled pLys.

### Redox-active coacervates
The redox active behavior of ferricyanide/ferrocyanide based coacervates were confirmed on a microplate reader (Tecan Spark) and followed by observed under microscope. Turbidity of a ferricyanide/pLys mixtures (ferricyanide 1 mM, Lys 5 mM, and with 50 mM tris buffer at pH = 7.4 or acetate buffer at pH = 4 to control pH) solution with a total starting volume of 200 μL above the critical salt concentration was monitored at a wavelength of 520 nm in 96-well plates

(Greiner Bio-one, clear flat-bottom wells). Adding 1 equiv. of NADH (2 µl, 100 mM stock solution) or 1 equiv. of GSH (2 µl, 100 mM stock solution) to the clear ferricyanide/pLys mixtures wells at pH 7.4 and pH 4 to study the pH effect between NADH and GSH, samples were shaken for 5 s before every readout and observed the samples under microscope. The redox cycles regulated by redox GSH and $S_2O_8^{2-}$ were performed by alternating additions of 1 equiv. of GSH (2 µl, 100 mM stock solution) and 0.5 equiv. of $S_2O_8^{2-}$ (1 µl, 100 mM stock solution), which show that condensation and dissolution are both reversible and that the system can be switched multiple times between a coacervates droplet state and a homogeneous solution.

### Peptide bond formation in ferricyanide/pLys coacervates

All the stock solution were prepared in $D_2O$, pLys(Me)₃ (200 mM), ferricyanide (200 mM), amino acids (AA, 200 mM), Ac-Gly-SH (100 mM). The mixing sequence is shown in Supplementary Fig. S38. The ferricyanide/pLys coacervates were formed first, followed by adding the AA substrates. The pH of the solution was adjusted to pH 9.0 with NaOD/DCl, after which Ac-Gly-SH was added to a total volume $V_t$ = 200 µL. The pH will decrease as the reaction progresses, which was monitored using pH paper. Additional aliquots of 1 M NaOD were added to ensure the pH of the reaction mixture was maintained at 9 ± 1, according to the diagram shown in Supplementary Fig. S38b. This strategy to maintain the pH was chosen to minimize the presence of ionic species solutes other than the coacervate components and the substrates of the ligation reaction in order to minimize the possibility of coacervate dissolution or the occurrence of side reactions. The ligation reactions were carried out for 60 or 72 h, with samples measured typically every 12 h. Each time point was prepared as a separate and independent sample. Before NMR analysis of a sample, 16 µL NaBH₄ (100 mM in $D_2O$) was added to quench the ferricyanide, and after that 184 µL NaCl (3 M in $D_2O$) was added to completely dissolve the coacervates. The final total volume was $V_t$ = 395–400 µL. NMR spectra were measured on a Bruker-AVANCE III 400 MHz spectrometer. As an alternative, the ligation reaction was carried out in 100 mM borate buffer (sodium borate, pH 9.2). No NaOD was used to adjust the pH during the reaction, and the pH was checked to remain constant throughout the ligation reaction. Everything else was kept the same as above.

### Fiber self-assembly inside ferricyanide-based coacervates

Water soluble $DBC^{2-}$ (20 mM, at pH 7) were firstly reduced with 1 equivalent of dithiothreitol (DTT) to give non-active monomer N-benzoyl-L-cysteine ($BC^-$). We flushed 20 µL $BC^-$ solution (20 mM, with 0.2 µL, 1 mg/mL Nile red) into a PLL-g-PEG modified microchamber containing 20 µL ferricyanide/polypeptides coacervates (8 mM ferricyanide, 24 mM polypeptides, monomer basis) at pH 7 as control, and at pH 3 for the fiber assembly inside the coacervates.

### Mass spectrometry

Mass spectra were obtained from a Thermo Scientific™ LCQ Fleet™ ion trap mass spectrometer with Gemini-NX C18 110 A 150 × 2.0 mm column and JEOL Accurate Time of Flight (ToF) instruments, both using linear ion trap electrospray ionization (ESI).

### Reporting summary

Further information on research design is available in the Nature Portfolio Reporting Summary linked to this article.

## Data availability

All data supporting the findings of this study are available within the article and in the Supplementary Information. Source data are deposited in the Radboud Data Repository (https://data.ru.nl) under accession code DOI 10.34973/k2n5-nb03. The data are available under CC-BY-NC license.

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

## Acknowledgements

This work was financially supported by the European Research Council (ERC) under grant number 851963 to E.S., and the Fundamental Research Funds for the Shanghai Sixth People's Hospital (X-2430 to J.W.). The authors would like to thank Dr. Karina Nakashima for synthesis of the *N*-acetyl glycine thioacid and helpful discussions about the project goals, Haibin Qian for preliminary experiments on EDC-mediated amide bond formation in pGlu/pLys(Me)₃ coacervates and the methylation protocol for pLys, Dr. Tiemei Lu for help with preparing the methylated pLys, and Wojciech Lipiński for help with the FRAP measurements and amino acid partitioning experiments.

## Author contributions

E.S. designed and supervised the project. J.Y.W. guided and supervised the project. J.H.W. designed, performed and analyzed peptide bond formation reactions and microscopy experiments. J.H.W. and M.A. carried out the synthesis and characterization of peptide bond formation reactions in coacervates. J.H.W. and E.S. wrote the manuscript, with input and revisions from all authors.

## Competing interests

The authors declare no competing interests.
