## [Peer Review File · Nature Communications]

REVIEWER COMMENTS

Reviewer #1 (Remarks to the Author):

The authors describe the effect of using peptide coacervates for concentrating ferricyanide, a potentially prebiotic oxidant, implicated in various experiments concerned with the origins of life. The peptide syntheses that were reported by Islam/Powner (ref. 7 of the manuscript) are shown to work within coacervate. The authors show ferricyanide concentrations are significantly higher within peptide coacervates than the external environment (the dilute phase), and this results in an increase in the rate of peptide ligation of an alpha-amidothioacid (Ac-AA-SH) and an amino acid within the peptide coacervate. The authors provide evidence of the oxidant localisation within the compartment by showing the formation of fibers within the coacervate. The confocal micrographs are showing these effects very clearly. There are a few things that the authors need to do before the MS is readied for as contribution in Nature Communications.

The amide bond forming reaction building on the work on ref. 7 is the centrepiece of the work though, but is introduced later in the paper. I would suggest that the manuscript be reordered so that the amide/peptide work come before the fiber studies with Bn-Cys-OH. The fiber formation does show that the oxidant is within the coacervate so despite my initial misgivings of that work being a mere 'add on' This is additional evidence of the redox activity within the coacervate. Do these fibers disappear upon addition of a reductant that will return the fibers back to Bn-Cys-OH?

The ligation seems to be much slower than the reported ligations in ref. 7. This might be because the pH may have dropped from 9 to a lower pH as the reaction progresses. I have mentioned this elsewhere in my comments below, and it is something the authors could check. The authors may find that the ligation works faster with an amine with a lower pKa, such as glycine nitrile as reported in ref. 7. It is interesting in Figure 4 that the reaction in the dilute phase is essentially already complete as soon as the reaction is monitored at the initial time point, whereas in the coacervate the product is gradually increasing over the course of days. Why is the reaction so slow within the coacervate? My only explanation can be that the pH has dropped.

I think further experiments are required with regards to the mixed amino acid experiments. We are told that glycine outcompetes Glu because Gly is 'more active'. The authors state that Glu may be 'bound more strongly to the cationic lysine residues inside the coacervate and therefore has a lower activity'. To test this hypothesis the authors could compare Gly and Asp – aspartate should therefore behave like Glu. Then, Gln (which is the amide of Glu) should produce more ligation product as it will 'more active' in the words of the authors, because it should not bind as strongly as Glu to the cationic lysine. Likewise for Asn. Asn has quite a low pKa compared to the other 19 amino acids. I'm not convinced yet by the rationale for the results that are being made. For example, this effect is apparently an 'example of

kinetic pathway selection'. Can we really draw that conclusion based on the limited data set one has at the moment? The data is currently insufficient in my opinion to make that conclusion. The matters are complicated then by the results with Ala and Phe in competition with Gly because Ala and Phe ligate better in the presence of Gly, but then we are told Glu outcompetes Phe. I think the authors should clarify what's going on here. The authors also mention that Ala and Phe have 'slightly stronger interaction with the peptide backbone in the coacervates' but I don't see any evidence for this in the manuscript. What is this 'interaction', and how is it working?

My major concerns have to be with the experiments related to Figure 5. The CSC of coacervates involving experiments with Ac-Phe-SH and Phe-SH are near-identical, and I would not have expected this because of the reported difference in the acylating behaviour of these two thioacids (see below). When the thioacid is oxidised in either Ac-Phe-SH or Phe-SH, and is then ligated (or hydrolysed by water to give the acid (CO₂H)), one of the by-products is the formation of polysulfides and inorganic sulfur (S₈), according to the proposed mechanism by Liu and Orgel (ref. 9 of the manuscript). Did the authors take into account that there will be sulfur precipitates produced when they carry out the experiments that begin on line 324 onwards? The authors are discussing turbidities of the coacervates at different salt concentrations, for example they describe that they observe 'gel-like droplets' that do not fuse and they have an increased salt resistance, which they attribute to acylation of the side chain epsilon amine of lysine. However, if Phe-SH acylated the epsilon amine of lysine one would expect there to remain a charge (the amine of the N-terminal Phe. This is why I am rather concerned especially about the experiments with Phe-SH. Already based on the experiments by Maurel and Orgel (ref. 6) of the manuscript (but also see <https://link.springer.com/content/pdf/10.1007/s11084-007-9070-9.pdf> - the authors should cite this too) we are told that the formation of amide bonds is not a very efficient reaction at all, especially in the absence of bicarbonate. There seems to be a need to use Glu10 to acylate Glu-SH. If one starts with Glu-SH (a monomer), the efficiency of aminoacylation is exceedingly poor. Therefore, how can the authors be sure that acylation with Phe-SH is occurring at the epsilon amines? The data from the two papers (ref. 6. And <https://link.springer.com/content/pdf/10.1007/s11084-007-9070-9.pdf> are awkward to read because they do not reveal the ligation yields or conversions to demonstrate how effective the acylations of Glu-SH are. This is why I am concerned that the 'gel-like' droplets may be an interference by inorganic/colloidal sulfur, especially in experiments with Phe-SH. NMR spectroscopy should reveal the extent of epsilon acylation by Phe-SH, in the way Ac-Phe-SH acylated polyLys, but the authors have not offered any data of experiments with PheSH except for the salt resistance studies in Figure 5b. I have looked at the supplementary figure 10, and the NMR relates to Ac-Gly-SH, not Ac-Phe-SH (I mentioned this again below, so apologies for the duplicate comment). Have the authors tried this reaction without coacervates? What does colloidal sulfur look in the microscopic images? Perhaps the authors could generate a mock-up mixture of the coacervate, sulfur, and ferri- and/or ferrocyanide and observe the effect. I am suggesting this because I don't want the authors to fall into a trap of making an observation that isn't really there. The acylation of the epsilon amine with Ac-Gly-SH is clear, but the reactions with Ac-Phe-SH and Phe-SH certainly aren't based on the data that I have at hand. Based on the implausibility of aminothioacids (e.g Phe-SH, Gly-SH etc) and their poor (amino)acylating potential it might be a good idea to consider whether these reactions should be even discussed in the context of this work. Much of

the work revolves around Ac-AA-SH and the ligation reaction reported in ref. 7. I will leave this up to the authors to think about and based on any further experiments they may carry out.

Minor

Names of molecule classes - Throughout the manuscript the authors refer to Ac-Gly-SH and related thioacids described in ref. 7 as 'aminothioacids'. They are not amino thioacids – Ac-Gly-SH and Ac-Phe-SH are alpha-amido thioacid (the alpha amine is an amide – hence 'amido'. If the thioacid was Ac-Gly-Gly-SH it too is an 'alpha-amido-thioacid' because the C-terminal thioacid's alpha position is an amide. On the other hand, Gly-SH and Phe-SH would be an amino thioacids. The distinctions between Ac-Gly-SH and Gly-SH are very important because both of these substrates behave entirely differently – Ac-Gly-SH is an electrophile and Gly-SH is both an electrophile and nucleophile. N-acetyl glycine (Ac-Gly-OH) and glycine (Gly-OH), for example, are not both amino acids. Gly-OH is an amino acid, whereas Ac-Gly-OH is not. Please could the authors go through the manuscript and make changes anywhere that this error occurs.

Abstract line 22- 'We demonstrate that aminoacylation is enhanced in Fe(CN)₆³⁻/peptide coacervate dispersions compared to the surrounding dilute phase, and selective for amino acids that interact less strongly with the coacervates.' Is it really possible to draw this conclusion from the limited data presented so far? Please see my comments about this in the mixed amino acid coupling experiments.

Line 31 "However, before ribosomes and specialized enzymes became capable of protein synthesis,^{1,2} alternative, simple prebiotic routes to create peptide bonds in a spatiotemporally controlled way likely existed." Why and what have to necessarily be spatiotemporally controlled?

Line 34 onwards "As plausible precursors to peptides, α -aminothioacids (AA-SH)⁶ and acetylated aminothioacids (Ac-AA-SH)⁷ have been shown to be formed in aqueous conditions at near-neutral pH." See comments about naming of AA-SH and Ac-AA-SH. Also it is important to make sure authors ensure the literature remains factual. Ref. 6 does not actually contain data for the synthesis of AA-SH in aqueous conditions. The thioacid is synthesised using conventional organic synthesis and then it's oligomerisation potential is investigated. The plausibility of AA-SH is clearly questioned from the literature, and formed an important discussion in ref. 7 (Canavelli et. al. Nature, 2019, where they wrote in the manuscript

"Maurel and Orgel have previously suggested that AA-SH¹⁶ (ref. 6 of this manuscript) might offer an interesting alternative to biological thioesters^{10,11}. AA-SH combine excellent aqueous stability with highly selective (electrophilic or oxidative) activation^{12,14,16,24}, but their prebiotic synthesis presents difficulties²⁵ (Leman, L. J. & Ghadiri, M. R. Synlett 28, 68–72 (2017)) and they undergo inefficient ligation at near-neutral pH (Supplementary Discussion)^{16,26}."

The authors of ref. 7. explain the problems of AA-SH plausibility and the problems of their polymerisation (which seem to only work in the presence of bicarbonate, and needs an oligopeptide primer to get the ligation to even proceed. The conditions of the synthesis of AA-SH in Leman, L. J. & Ghadiri, M. R. *Synlett* 28, 68–72 (2017) are not productive for AA-SH synthesis., which also produces CO₂/bicarbonate, also decomposes the amino thioacid.

Line 41 – “However, high concentrations of reactants and catalysts are typically required for these oxidative peptide ligations, which may not have been easy to reach.” - In ref. 7. the authors demonstrate that a related oxidative coupling of an alpha-amidothioacid (Ac-Val-SH) and glycine nitrile (Gly-CN) with ferricyanide can work at concentrations as low as 0.5 mM Ac-Val-SH and 1.5 mM in Gly-CN and ferricyanide. These are very, very low concentrations and they do show that the peptide synthesis is significantly slower (>24h to go to completion) The data is in their supplementary information. This does not compromise this present manuscript at all because in the present case we are seeing the relative effect within and outside the coacervate. Therefore, there is a significant relative rate enhancements caused by compartmentalisation. Can the authors determine what the pH of the dilute phase (outside the coacervate) and within the coacervate at the start of the peptide synthesis reaction, and as the reaction progresses? Presumably the pH is dropping as the peptide synthesis proceeds. The authors could try the reaction with an aminonitrile instead of glycine. The authors may find that the reaction works better using Gly-CN as their nucleophile because of its significantly lower pKa.

Line 48 – ‘readily’ instead of ‘easily’? Also ‘guest molecule’ – it’s not a guest molecule until it is inside the coacervate.

Line 52 – ‘overall rate’ of...?

Line 71 – ‘possible to make coacervates’ – use produce or synthesise instead of ‘make’?

Line 77 onwards- ‘Alternative oxidizing agents, such as ferricyanide (Fe(CN)₆³⁻), could have has been essential in the prebiotic activation of building blocks for peptide ligation,7,9,33,34

The authors have cited ref. 33 on the discussion of ferrocyanide, presumably as a plausible prebiotic substrate. Alongside it they should cite more recent discussions on the formations of ferri- and ferrocyanides, including

<https://www.sciencedirect.com/science/article/pii/S0016703719303801>

<https://www.science.org/doi/epdf/10.1126/sciadv.aax3419>

Line 81 – ‘ferricyanide-catalyzed oxidation reactions’ – are there any ferricyanide-catalyzed oxidation reactions in the manuscript? Should we instead say ‘ferricyanide-mediated?’ Unless I have missed an catalytic oxidations in the manuscript.

Line 96 – ‘client substrate’ – what’s that?

Line 117 – what do we mean by ‘salt-free’ solution? I think we mean NaCl-free in these cases. The reason why I am a little pedantic about this is that ferrocyanide and ferricyanide are, by definition, salts.

Line 119- “Increasing the length of polylysine to (Lys)₃₀ resulted in the formation of coacervates with trivalent ferricyanide as well (Supplementary Fig. 4c), in agreement with previous studies involving nucleotides.^{19,27} “

What is the reason for ferricyanide now working with longer chains? I think the authors should state this specifically as to how and why this is ‘in agreement with previous studies involving nucleotides’.

Line 131- ‘For example, for samples prepared from 2 mM ferricyanide and pLys (5mM lysine monomers) solutions, we found an internal ferricyanide concentration of ~30 mM, compared to ~0.3 mM in the surrounding aqueous phase. (Supplementary Fig. 5).’ – how long was this concentration maintained within the coacervate? Can the concentrations be modified? For example if you started with 1 mM ferricyanide what are the concentrations within and outside the coacervate, and does addition of ferricyanide to the bulk solution lead to a readjustment of concentrations in the dilute and within coacervate?

Line 161 – ‘From plots of the turbidity we determined the critical salt concentration (CSC), the point at which coacervate droplets completely disappear.’ Can we state specifically what number of the CSC is?

Line 167- ‘we prepared mixtures of Fe(CN)₆⁴⁻ with pLys and Fe(CN)₆³⁻ with pLys under identical conditions within the highlighted region of Fig. 1g between the two binodal lines.’ – what are the specific ‘identical conditions’ within the highlighted regions? Can we report concentrations here, or in the SI?

Figure 2. Why is it that NADH and GSH work at directly opposite pH? Is there a stability issue with the molecules at pH where they don’t work? Instead of GSH have the others tried a different thiol e.g N-acetylcysteine or DTT? I am not asking for further experiments here, I am just curious.

Line 265- 'Ferricyanide has been described as a prebiotically abundant oxidizing agent,33

Please add <https://www.sciencedirect.com/science/article/pii/S0016703719303801>

<https://www.science.org/doi/epdf/10.1126/sciadv.aax3419>

Line 265 '...and has been used to activate amino thioacids by oxidation to facilitate the formation of an amide bond upon reaction with nucleophilic aminonitriles and amino acids.6– 9,34'

Ref 8. does not use ferricyanide. Ref. 34 does not use amino thioacids. It uses the ferricyanide to oxidise the thiocarbamate that results from the reaction of an amino acid and carbonyl sulfide. This can then cyclise to form an N-carboxyanhydride. The acylating agent is an N-carboxyanhydride.

Line 272. 'However, the oxidative aminoacylation of thioacids is usually performed with high reactant and ferricyanide concentrations (sometimes close to 100 mM), which could have been difficult to reach everywhere on Early Earth.' See comments above about low concentration oxidative peptide ligation in ref. 7. As I said, that does not detract from this work because it's showing the relative rates of ligation within and external to the coacervate. However, the authors should not make the reader think there is a problem that needs to be resolved (low concentration of oxidant for oxidative ligation), rather, the author's manuscript is showing a physicochemical phenomena generated by the coacervate. That is what you should be stressing in your manuscript, which I feel you haven't done forcefully yet but should do in revision.

Generally throughout manuscript – please check carefully where you have used which peptide for coacervation. Eg Supplementary Fig. 10 – did you use Arg10 or pLys?

Supplementary Information

I would recommend that the authors spend time in writing out experimental protocols that reflect what the experimenter did for all experiments discussed within the text and in the figures. I think this should be done as standard at all times. It also makes the experimenter think again whether the procedure they carried out was correctly executed when writing this all up.

Data in figures:

Figure 1. The experiments with NADH, polyu-rU15 and pyranine are not described within the experimental section. The procedures are in the SI right now are so highly generalised. Quantities could be reported so as to reproduce this work. For example, as the reviewer, I cannot assess whether the quantities used in experiments are correct (or not).

Figure 2. These experimental procedures are not described as far as I am aware.

Figure 3. Again, not described.

Figure 4. Not described.

Line 339 'In contrast, when Ac-Phe-SH/Phe-SH was added and reacted with the ϵ -NH₂ of pLys, the turbidity transition shifted to a significantly higher salt concentration and the absolute intensity of the plateau at high salt concentration was higher (Fig. 5b).' this reads as Ac-Phe-SH and Phe-SH in the same reaction – I don't think it is. A description of these experiments would resolve ambiguity in the SI.

Supplementary Table 1 – The selectivity ratios are reported, but what are the actual conversions/yield of ligations?

Supplementary Figure 12 – Can we get a number on how much epsilon amine was acylated? Also, the figure shows Ac-Gly-SH, but the text refers to experiments with Ac-Phe-SH.

Good luck with the revisions, and I look forward to seeing the manuscript again.

Reviewer #2 (Remarks to the Author):

Peptide-based coacervates have been widely used as model protocells in the field, however the synthesis of peptides inside of them has not been demonstrated. This is an outstanding problem that the authors attempt to address in this study using molecules that may have been readily available on the primitive Earth. Here they use ferricyanide (Fe³⁺) and Lysine/arginine coacervates as redox active coacervates. Impressively, they demonstrate that 2mer's made from amino thioacids and peptides/amino acids can be made with fairly high yield by oxidizing the thioacid. Overall this represents a fairly big increase in the field because it presents a chemically feasible way to achieve peptides before invention of the ribosome.

There are a number of positive features of this study. For instance, the authors demonstrate that coacervates made from ferri/ferrocyanide PLUS pLys can partition small molecules, which can be used to tune the oxidation state and the coacervation. Moreover this process can be cycled multiple times indicating that oxidation and reduction by alternating additions of GSH and S₂O₈²⁻ is reversible, as is

formation of compartments and homogeneous solution of one phase. Additionally, the authors demonstrate formation of peptide bonds in side of the droplets and that this is selective dependent on the amino acids/amino thioacids involved. Finally, the authors demonstrate novelty in that when hydrophobic amino acids are incorporated the properties of the droplets change.

Major Comments:

1. While this article is about peptides, it is important to think about compatibility with RNA. The authors should test if their coacervate droplets are friendly to RNA or if they degrade the RNA. This will increase the impact and scope of the work if successful
2. The authors make claims about the liquidity of the droplets in figure 1, in Supplementary figure 8, and in Figure 5. Their observations about wetting of surfaces and coalescence of droplets are good, but FRAP experiments could be done to further demonstrate whether droplets are liquid or gel-like.
3. Some discussion of what droplets could do which things would be appreciated. In figure 3 they focus on arg droplets, but for all the rest of the paper it is pLys and its derivatives. Did the experiments not work for the arg droplets?
4. The statement on line 202 says that after 8 cycles the droplets can't be reformed, could this also be because of dilution of the coacervate components and not just of product build up?
5. Line 216 They make a statement about fluorescence disappearing from the center of the droplet outward, if possible they should quantitate this phenomenon as it is important to their point.
6. In the figure 3 legend, they describe a time series, but don't make it clear on the figure whether microscope images come from different times. Please clarify this.
7. In Supplementary Figure 4h they should include another panel with a bright field image of those same droplets. Also why are there strange lines in supplementary figure 4f?
8. In supplementary figure 8c, the droplets appear to be solid aggregates, they should discuss this.
9. Line 284, did the researchers do the peptide bond experiment in ferrocyanide-pLys(Me)₃ coacervates? As this would be a direct comparison to their other experiment in ferricyanide-pLys(Me)₃. If this doesn't form a phase, it makes sense, but they should state that.
10. Line 316 they make a statement about kinetic pathway selection when there is Gly and Glu there. Some discussion of how this compares to Glu only would be helpful here.
 - a. Also some mention of the N of experiments would be good. If N is 1, suggest completing at least 2 more replicates.
11. In general, they should give the # of experiments in the figure legends for all of the experiments that they did.
 - a. This is particularly lacking in figures 2, 4, and 5

12. Figure 5C they state these droplets are gels, consider doing FRAP to demonstrate this. Additionally, some type of statistics should be given to know that this image is representative of the entire field.

13. Supplementary Figure 6, it would be helpful to have some quantitation of the turbidity. e.g. See Fig 3 in reference 19.

Minor comments

1. They should be consistent about use of the short form of ferri/ferrocyanide. Seems random now.

2. In figure 3 or in the text some discussion of the time required to form fibrils would be good.

3. Line 282 be specific about which coacervates. There are many described in the paper and the next line there is talk of a different coacervate.

4. Line 290 it would give the readers more context to either give the mass of the shorter Lys derivatives or give the dispersion of sizes of the pLys because their point is that the short ones do the reaction better.

5. In Figure 4, they show formation of a 2mer inside of the coacervates. Is their evidence that longer peptides can be made given addition of more substrates?

6. State what GSH means in Figure 5 figure legend.

7. Supplementary Information should have a table of contents at the beginning to make it clear what is where in the SI.

Reviewer #3 (Remarks to the Author):

In this manuscript, Jiahua Wang et al. utilize coacervate droplets made from redox-active ferricyanide (Fe^{3+}) or ferrocyanide (Fe^{2+}) and polylysines or oligoarginine as protocell models. With the addition of a reducing or oxidizing agent, they can switch between the phase-separated and the mixed state for multiple cycles. Interestingly, both assembly and disassembly can be performed at either low or neutral pH, dependent on the reducing agent. They show that they can use the oxidizing properties of ferricyanide to trigger fiber assembly inside the droplets via oxidation of a benzoyl thiol. Finally, they use the propensity of ferricyanide to accelerate oxidative aminoacylations of thioacids to dimerize compartmentalized amino acids. They show that some amino acid dimers are selectively formed in favor of others and they claim that this is due to different binding affinities to the droplet's building blocks. In some cases, the droplet building blocks themselves are modified so that the resulting compartment becomes more salt-resistant which the authors propose as an advantage in terms of protocell fitness.

In general, the results shown here are promising and could be of great value in the origin of life field. Most reported compartmentalized reactions in coacervate droplets focus on enzymes or ribozymes but synthesis of peptide building blocks is indeed scarce. Also, in terms of redox-active coacervates as protocell models little is reported so far. I do, however, have several major and minor concerns:

1. Mechanism of selective amino acid dimerization. The authors argue that glycine is the most reactive amino acid since it has the lowest affinity to polylysine, one of the two main droplet building blocks. If so, it seems counter-intuitive to me that glutamic acid with its negative charges is more reactive than phenylalanine, which in turn should have weaker interactions with the polycation than glutamate. Also, using only glutamic acid, the dimer Gly-Glu is formed with comparable rates than Gly-Gly (in case of 100% Gly), indicating that the polylysine interaction is not decisive here. Therefore, the proposed mechanism seems to be rather vague and not fully explain what is going on. I wonder whether other parameters are much more important, eg. sterically hinderance. One could test this maybe with other bulky amino acids? Ideally one could use basic amino acids such as lysine or arginine to have a control for the glutamic acid. But I assume that the side chain would react as well so methylated lysine or arginine would be needed? One could also try to modulate the affinity towards glutamic acid by lowering the pH or using a protected derivative.
2. Enhancement of ligation in droplets with shorter peptides. The authors attribute this effect to the lower multivalency of the polycation, which in turn would result in weaker complexation of the ferricyanide. What about other effects such as diffusivity of the molecules inside of the droplet which is directly linked to the length of the polymers. Techniques like FRAP would help here.
3. Protocell fitness. The authors argue that the salt-resistance from side-rctns that modify the droplet's main building blocks may be advantageous in terms of protocell fitness. I would agree with that until a certain point. If the metabolism inside a protocell gets out of control, one will not get a fitter compartment but just an agglomerate that has lost its function. One of the reasons why coacervates are regarded as relevant protocell models is because they have considerable water inside, so the molecules inside the droplet can move rapidly. Again, FRAP would be a good first indication of how fluid the droplets still are. Also, it would be nice to see the new CSC of the "fitter" droplets since it is not shown in Fig 5b.
4. Reversibility. In Figure 2, the authors explain that one can switch between assembly and disassembly via reduction of ferricyanide and oxidation of ferrocyanide. However, in Figure 5b and Supplementary Figure 7, the reduction of ferricyanide to ferrocyanide dissolves the droplets. This is not intuitive because of the increase in charge density and valency. Something seems off.

Minor comments:

1. Since there are no microscopy images in Figure 4. Are the droplets affected in any way by the reaction inside?
2. In S7, the fluorescent "holes" appear due to an oxidation rctn. In Fig 1f and S4, however, these holes are also present. Why?

3. In S9&10, spectra of later time points would be good, similar as in Fig 4b. Also, the peaks in S9&10 are slightly shifted. Is that due to different deuterated solvents?
4. Figure 5b: caption misses what type of coacervates
5. In S12, the ligation of glycine to polylysine is shown. However, in the text and in Figure 5, the ligation of phenylalanine is discussed. Also, should not there be more orange proton peaks?

Reviewer #1 (Remarks to the Author):

1. The authors describe the effect of using peptide coacervates for concentrating ferricyanide, a potentially prebiotic oxidant, implicated in various experiments concerned with the origins of life. The peptide syntheses that were reported by Islam/Powner (ref. 7 of the manuscript) are shown to work within coacervate. The authors show ferricyanide concentrations are significantly higher within peptide coacervates than the external environment (the dilute phase), and this results in an increase in the rate of peptide ligation of an alpha-amidothioacid (Ac-AA-SH) and an amino acid within the peptide coacervate. The authors provide evidence of the oxidant localisation within the compartment by showing the formation of fibers within the coacervate. The confocal micrographs are showing these effects very clearly. There are a few things that the authors need to do before the MS is readied for as contribution in Nature Communications.

The amide bond forming reaction building on the work on ref. 7 is the centre piece of the work though, but is introduced later in the paper. I would suggest that the manuscript be reordered so that the amide/peptide work come before the fiber studies with Bn-Cys-OH. The fiber formation does show that the oxidant is within the coacervate so despite my initial misgivings of that work being a mere 'add on' This is additional evidence of the redox activity within the coacervate. Do these fibers disappear upon addition of a reductant that will return the fibers back to Bn-Cys-OH?

► We thank the reviewer for their thorough feedback on our manuscript. We have reordered the manuscript as suggested. With respect to the question on reversibility of the fibers, these are reversible upon addition of reducing agent, such as β -mercaptoethanol, as we show in Supplementary figure 35-36. These findings are in agreement with the original report by Wojciechowski et al., JACS 2018 (ref. 53 in our manuscript).

2. The ligation seems to be much slower than the reported ligations in ref. 7. This might be because the pH may have dropped from 9 to a lower pH as the reaction progresses. I have mentioned this elsewhere in my comments below, and it is something the authors could check. The authors may find that the ligation works faster with an amine with a lower pKa, such as glycine nitrile as reported in ref. 7. It is interesting in Figure 4 that the reaction in the dilute phase is essentially already complete as soon as the reaction is monitored at the initial time point, whereas in the coacervate the product is gradually increasing over the course of days. Why is the reaction so slow within the coacervate? My only explanation can be that the pH has dropped.

► We thank the reviewer for this suggestion. The pH of the reaction was kept constant at pH 9 (± 1) throughout the experiment by dropwise addition of 1 M NaOD. The question raised by the reviewer made us realize that this aspect was not described sufficiently clearly in our methods, and we have now updated the methods section and added a trace of base addition during the experiment, which provides further insight into the reaction progress. To prove that the reaction rate is not caused by a decrease of the pH, we have now repeated the experiments in borate buffer (Supplementary Fig. 15), and we see the same rate of ligation.

Our motivation for performing the reactions initially without buffer is that we wanted to keep the system as simple as possible, leaving out any unnecessary ions, which might either destabilize the

coacervates or interfere with the oxidation reaction, as our main objective was to demonstrate a significant effect of ferricyanide compartmentalization by coacervates on the ligation reaction. For the control experiments performed with borate buffer, we have checked that the stability of the coacervates is not affected.

The reason for the lower rate in our work compared to the work in ref. 7 is the much higher pKa of glycine that we used compared to glycine nitrile used in ref. 7, and the lower concentrations of ferricyanide, especially in the supernatant outside of the coacervates. At pH 9, the amine of glycine will be mostly protonated, in contrast to glycine nitrile, and therefore the effective concentration of reactive species is significantly lower. Nevertheless, we decided to use amino acids rather than aminonitriles, to show that coacervates could even promote the ligation of amino(thio)acids. Aminonitriles are much more reactive but also less stable than amino acids. We reasoned that the ligation of the more stable amino acids, even if this would proceed at slower rate, would be an interesting achievement from a prebiotic perspective, in particular if the yields are similar to the yields of aminonitriles.

The reviewer also asked about the apparent completion of the reaction in the dilute phase. We would like to argue that the reaction in the dilute phase is much slower than in the coacervate phase, and certainly not completed as soon as the reaction is monitored. We believe that the small signal detected at $t=2\text{h}$ (our first measurement point) may be due to slightly overlapping peaks of the added peptide or a small impurity rather than actual product formation (we monitor the product formation between 2.0 and 2.2 ppm). The signal does not increase significantly in the dilute phase, as shown in the spectra below, while a clear peak emerges in the coacervate dispersion, as shown for example in Figure 3b. To avoid confusion about the completion of this reaction, we have now subtracted the signal in a control sample with ferrocyanide, in which the reaction does not proceed.

Figure R1. NMR analysis of the reaction between Ac-Gly-SH and Gly (left) or Glu (right) in the presence of ferricyanide in the dilute phase showing that the peaks corresponding to the product do not increase over time.

3. I think further experiments are required with regards to the mixed amino acid experiments. We are told that glycine outcompetes Glu because Gly is 'more active'. The authors state that Glu may be 'bound more strongly to the cationic lysine residues inside the coacervate and therefore has a lower activity'. To test this hypothesis the authors could compare Gly and Asp – aspartate should therefore behave like Glu. Then, Gln (which is the amide of Glu) should produce more ligation product as it will 'more active' in the words of the authors, because it should not bind as strongly as Glu to the cationic lysine. Likewise for Asn. Asn has quite a low pKa compared to the other 19 amino acids. I'm not convinced yet by the rationale for the results that are being made. For example, this effect is apparently an 'example of kinetic pathway selection'. Can we really draw that conclusion based on the limited data set one has at the moment? The data is currently insufficient in my opinion to make that conclusion. The matters are complicated then by the results with Ala and Phe in competition with Gly because Ala and Phe ligate better in the presence of Gly, but then we are told Glu outcompetes Phe. I think the authors should clarify what's going on here. The authors also mention that Ala and Phe have 'slightly stronger interaction with the peptide backbone in the coacervates' but I don't see any evidence for this in the manuscript. What is this 'interaction', and how is it working?

► To further support our conclusions, we have measured the partitioning coefficients of the amino acids used in Figure 3 and Table 1, and we included additional experiments with mixed amino acids using Asp, Asn and Gln in combination with either Gly, Glu, Ala or Phe. The results are presented in Supplementary fig. 21-32, and discussed in our manuscript in relation to the discussion of Figure 3d (in the revised manuscript).

The partitioning coefficient reflects the strength of the noncovalent interactions (charge-charge, cation- π , π - π , hydrogen bonding) between the amino acids and the other coacervate components. The same interactions underlie the differential partitioning of small molecules, such as NADPH (Supplementary Fig. 11) into the coacervates. Stronger partitioning means higher local concentrations, but it also leads to decreased “activity” inside the coacervate. Recent theoretical work supports our hypothesis that stronger interactions lead to decreased reaction fluxes in phase separated systems (Bauermann et al., JACS 2022, <https://pubs.acs.org/doi/epdf/10.1021/jacs.2c06265>). The results of the additional experiments we did with other amino acids are also in qualitative agreement with this principle.

We agree with the reviewer that the interpretation of relative yields of the ligation products is complex and very likely involves more than interactions with the coacervate. The pKa of the amino acids also plays a role and we have added a remark about the role of pKa and other factors to the discussion. However, the pKa of most of the studied amino acids (Gly, Ala, Glu and Asp) are very similar, and their yields when added separately to coacervates are indeed similar (Figure 3c-d, and Table 1), while their yields in experiments with mixed amino acids differ significantly. We therefore believe that the significant difference in products using mixed amino acids must be a result of the non-covalent interactions of the amino acids with the coacervate.

4. My major concerns have to be with the experiments related to Figure 5. The CSC of coacervates involving experiments with Ac-Phe-SH and Phe-SH are near-identical, and I would not have expected this because of the reported difference in the acylating behaviour of these two thioacids (see below). When the thioacid is oxidised in either Ac-Phe-SH or Phe-SH, and is then ligated (or hydrolysed by water to give the acid (CO₂H)), one of the by-products is the formation of polysulfides and inorganic sulfur (S₈), according to the proposed mechanism by Liu and Orgel (ref. 9 of the manuscript). Did the authors take into account that there will be sulfur precipitates produced when they carry out the experiments that begin on line 324 onwards? The authors are discussing turbidities of the coacervates at different salt concentrations, for example they describe that they observe 'gel-like droplets' that do not fuse and they have an increased salt resistance, which they attribute to acylation of the side chain epsilon amine of lysine. However, if Phe-SH acylated the epsilon amine of lysine one would expect there to remain a charge (the amine of the N-terminal Phe. This is why I am rather concerned especially about the experiments with Phe-SH. Already based on the experiments by Maurel and Orgel (ref. 6) of the manuscript (but also see <https://link.springer.com/content/pdf/10.1007/s11084-007-9070-9.pdf> - the authors should cite this too) we are told that the formation of amide bonds is not a very efficient reaction at all, especially in the absence of bicarbonate. There seems to be a need to use Glu10 to acylate Glu-SH. If one starts with Glu-SH (a monomer), the efficiency of aminoacylation is exceedingly poor. Therefore, how can the authors be sure that acylation with Phe-SH is occurring at the epsilon amines? The data from the two papers (ref. 6. And <https://link.springer.com/content/pdf/10.1007/s11084-007-9070-9.pdf> are awkward to read because they do not reveal the ligation yields or conversions to demonstrate how effective the acylations of Glu-SH are. This is why I am concerned that the 'gel-like' droplets may be an interference by inorganic/colloidal sulfur, especially in experiments with Phe-SH. NMR spectroscopy should reveal the extent of epsilon acylation by Phe-SH, in the way Ac-Phe-SH acylated polyLys, but the authors have not offered any data of experiments with PheSH except for the salt resistance studies in Figure 5b. I have looked at the supplementary figure 10, and the NMR relates to Ac-Gly-SH, not Ac-Phe-SH (I mentioned this again below, so apologies for the duplicate comment). Have the authors tried this reaction without coacervates? What does colloidal sulfur look in the microscopic images? Perhaps the authors could generate a mock-up mixture of the coacervate, sulfur, and ferri- and/or ferrocyanide and observe the effect. I am suggesting this because I don't want the authors to fall into a trap of making an observation that isn't really there. The acylation of the epsilon amine with Ac-Gly-SH is clear, but the reactions with Ac-Phe-SH and Phe-SH certainly aren't based on the data that I have at hand. Based on the implausibility of amino thioacids (e.g Phe-SH, Gly-SH etc) and their poor (amino)acylating potential it might be a good idea to consider whether these reactions should be even discussed in the context of this work. Much of the work revolves around Ac-AA-SH and the ligation reaction reported in ref. 7. I will leave this up to the authors to think about and based on any further experiments they may carry out.

► We appreciate the reviewer's concerns about the experiments in Figure 5 and thank them for all the suggestions, which motivated us to take a careful look at these experiments, perform additional control experiments, and re-evaluate the combined results.

Firstly, we did take into account the formation of sulfur precipitates in these reactions. In fact, the sulfur precipitates are clearly visible when observing the samples under a microscope, and very distinct from the coacervate droplets. A typical microscopy image of the samples prepared in Figure 5 showing the appearance of colloidal sulfur inside and outside coacervate droplets and intact coacervate droplets is shown below. Note that these colloidal sulfur clusters are also visible in the microscope images in Figure 5c in the original manuscript.

Figure R2. Microscope image of reaction mixture of ferricyanide (8 mM)/pLys (15-30K, 24 mM) with Ac-Phe-SH.

It is interesting to see that a notable fraction of the colloidal sulfur appears inside the coacervate droplets, which suggests that the ligation reaction in which it is produced occurs inside the droplets. When increasing the ionic strength to above the CSC, the coacervates are dissolved, but the colloidal sulfur remains, which explains why the turbidity plateau at high salt concentration of samples that have been incubated with Ac-Phe-SH and Phe-SH is higher than the controls beyond CSC. However, the colloidal sulfur cannot explain the shift in critical salt concentration.

We agree with the reviewer that the pKa of the epsilon-amine of polylysine is significantly higher than the alpha-amines of most amino acids. It will therefore be mostly protonated under the conditions used in our experiments, and the ligation yield is expected to be significantly lower. Nevertheless, we observed clear ligation with Ac-Gly-SH (NMR data shown in original Figure S12). The concentration of alpha-amino groups in this sample with a 15-30 kDa polylysine polymer is only 0.1-0.2 mM and could not explain the ligation peak area. Controls with polylysine alone, Ac-Gly-SH alone and Ac-Gly-SH with ferricyanide did not show this same peak. This ligation product must therefore be from epsilon-amine ligation. Based on this result and the fact that Ac-Gly-SH and Ac-Phe-SH have similar pKa and ligation yield with alpha-amino acids, we reasoned that also Ac-Phe-SH (and Phe-SH) could undergo ligation with the epsilon-amines. Moreover, weak modification of the epsilon-amines of polylysine could already be sufficient to observe a shift in CSC and the formation of gel-like coacervates. Other reports in literature indeed show that liquid-liquid phase separation and liquid-to-solid transitions can be triggered or suppressed by the slightest modifications to interactions. Gabryelczyk et al. (<https://doi.org/10.1038/s41467-019-13469-8>) show that a 23-mer peptide rich in His, Tyr and Phe loses its ability to phase separate upon modifying a single Tyr to Ala. Nott et al. (<http://dx.doi.org/10.1016/j.molcel.2015.01.013>) show that methylating 9 Arg residues on a 26 kDa protein (which also does not change the net charge) shifts the CSC of the protein from 200 to 100 mM at the same temperature. Finally, conversion of a single Gly to Glu in a disordered region

of FUS has been shown to induce the transition from a liquid to a solid-like phase (<https://doi.org/10.1016/j.cell.2015.07.047>), while replacing Tyr for Ser in FUS completely prevents its phase separation (<https://doi.org/10.1016/j.cell.2018.06.006>).

To further prove that ligation of Ac-Phe-SH and Phe-SH to the epsilon amines of lysine is possible, we tried to detect the ligation product by NMR and LC-MS. However, despite several attempts, including at higher concentration and pH (up to pH 10), we were unsuccessful in detecting any ligation product. We could detect ligation of Ac-Phe-SH and Boc-Lys-OH in solution at pH 9 in the presence of 100 mM ferricyanide (approximately the concentration inside coacervates) by LC-MS, as shown in the figure below. No significant signal was observed by NMR, suggesting that the ligation yield is quite low.

Figure R3. LC-MS analysis of the reaction mixture of Ac-Phe-SH and Boc-Lys-OH and controls with only Ac-Phe-SH and only Boc-Lys-SH to investigate the possibility of ligation at the epsilon amine of lysine.

Taking everything together, we did not find direct evidence of the ligation between Ac-Phe-SH and epsilon-amines in polylysine, which was our explanation for the shift in CSC observed in Figure 5. While it is still possible that the ligation yield is simply very low, but still sufficient to cause a shift of the CSC (which is not unlikely in light of the literature reports discussed above), we have to conclude that we cannot be certain of this explanation and we therefore decided it is better to remove this experiment from the revised manuscript. We have reorganized the manuscript (see also comment 1 by Reviewer 1) and discuss the fiber formation by Bn-Cys-OH inside coacervates now as the last topic, which shows that the localized oxidation potential of ferricyanide can be used to control the appearance and properties of coacervate protocells. Moreover, we believe the additional experiments (FRAP, mixed amino acids, partitioning) have further strengthened the manuscript. Removal of the original Figure 5 therefore does not alter the conclusions or the relevance of our work.

Minor

5. Names of molecule classes - Throughout the manuscript the authors refer to Ac-Gly-SH and related thioacids described in ref. 7 as 'aminothioacids'. They are not amino thioacids – Ac-Gly-SH and Ac-Phe-SH are alpha-amido thioacid (the alpha amine is an amide – hence 'amido'. If the thioacid was Ac-Gly-Gly-SH it too is an 'alpha-amido-thioacid' because the C-terminal thioacid's alpha position is an amide. On the other hand, Gly-SH and Phe-SH would be an amino thioacids The distinctions between Ac-Gly-SH and Gly-SH are very important because both of these substrates behave entirely differently – Ac-Gly-SH is an electrophile and Gly-SH is both an electrophile and nucleophile. N-acetyl glycine (Ac-Gly-OH) and glycine (Gly-OH), for example, are not both amino acids. Gly-OH is an amino acid, whereas Ac-Gly-OH is not. Please could the authors go through the manuscript and make changes anywhere that this error occurs.

- ▶ We have revised all compound names in the manuscript according to the reviewer's comments.

6. Abstract line 22- 'We demonstrate that aminoacylation is enhanced in Fe(CN)₆/peptide coacervate dispersions compared to the surrounding dilute phase, and selective for amino acids that interact less strongly with the coacervates.' Is it really possible to draw this conclusion from the limited data presented so far? Please see my comments about this in the mixed amino acid coupling experiments.

- ▶ We refer to our response to point 3. We have included additional experiments with Gln, Asn and Asp and partitioning data to provide further support to this conclusion. As the ratio of ligation products may also depend on other parameters, as also discussed above, we have reworded this phrase in the abstract and conclusion, to better express the fact that it is difficult to draw a conclusion about amino acids interacting with the coacervates without having a quantitative measure of the interaction.

7. Line 31 "However, before ribosomes and specialized enzymes became capable of protein synthesis, 1,2 alternative, simple prebiotic routes to create peptide bonds in a spatiotemporally controlled way likely existed." Why and what have to necessarily be spatiotemporally controlled?

- ▶ Ribosomes and enzymes are largely composed of proteins containing peptide bonds, which had to be formed in some way before (modern) ribosomes could form them. It is a popular belief that ribozymes could have played an important role, or that they were formed by condensation in wet-dry cycles. At some point, cell-like entities were present and peptides were also formed inside these compartments – this is what we meant with spatiotemporal control. Without compartmentalization, the ligation products would likely not have been able to accumulate to create more complex molecular structures.

8. Line 34 onwards "As plausible precursors to peptides, α -aminothioacids (AA-SH)⁶ and acetylated aminothioacids (Ac-AA-SH)⁷ have been shown to be formed in aqueous conditions at near-neutral pH." See comments about naming of AA-SH and Ac-AA-SH. Also it is important to make sure authors ensure the literature remains factual. Ref. 6 does not actually contain data for the synthesis of AA-SH in aqueous conditions. The thioacid is synthesised using conventional organic synthesis and then it's oligomerisation potential is investigated. The plausibility of AA-SH is clearly questioned from the literature, and formed an important discussion in ref. 7 (Canavelli et. al. Nature, 2019, where they wrote in the manuscript "Maurel and Orgel have previously suggested that AA-SH¹⁶ (ref. 6 of this manuscript) might offer an interesting alternative to biological thioesters^{10,11}. AA-SH combine excellent aqueous stability with highly selective (electrophilic or oxidative) activation^{12,14,16,24}, but their prebiotic synthesis presents difficulties²⁵

(Leman, L. J. & Ghadiri, M. R. Synlett 28, 68–72 (2017)) and they undergo inefficient ligation at near-neutral pH (Supplementary Discussion)16,26."

The authors of ref. 7. explain the problems of AA-SH plausibility and the problems of their polymerisation (which seem to only work in the presence of bicarbonate, and needs an oligopeptide primer to get the ligation to even proceed. The conditions of the synthesis of AA-SH in Leman, L. J. & Ghadiri, M. R. Synlett 28, 68–72 (2017) are not productive for AA-SH synthesis., which also produces CO₂/bicarbonate, also decomposes the amino thioacid.

► We have rephrased this sentence and added a sentence to highlight the discussion in literature about the plausibility of AA-SH.

9. Line 41 – "However, high concentrations of reactants and catalysts are typically required for these oxidative peptide ligations, which may not have been easy to reach." - In ref. 7. the authors demonstrate that a related oxidative coupling of an alpha-amidothioacid (Ac-Val-SH) and glycine nitrile (Gly-CN) with ferricyanide can work at concentrations as low as 0.5 mM Ac-Val-SH and 1.5 mM in Gly-CN and ferricyanide. These are very, very low concentrations and they do show that the peptide synthesis is significantly slower (>24h to go to completion) The data is in their supplementary information. This does not compromise this present manuscript at all because in the present case we are seeing the relative effect within and outside the coacervate. Therefore, there is a significant relative rate enhancements caused by compartmentalisation. Can the authors determine what the pH of the dilute phase (outside the coacervate) and within the coacervate at the start of the peptide synthesis reaction, and as the reaction progresses? Presumably the pH is dropping as the peptide synthesis proceeds. The authors could try the reaction with an aminonitrile instead of glycine. The authors may find that the reaction works better using Gly-CN as their nucleophile because of its significantly lower pka.

► We refer to our response to point 2 above – we used NaOD addition to maintain the pH at 9, and have now repeated the experiments with borate buffer with the same results. Because coacervates lack a membrane and they are permeable to all small molecules, the pH inside and outside is likely identical or very similar. The only report in literature in which a distinct pH value has been reported inside coacervates was by Keating and co-workers (<https://doi.org/10.1038/s41467-020-19775-w>). However, they found that the pH was not significantly different for polylysine of 30 residues or more. We therefore expect no significant difference in pH inside and outside the coacervates.

We understand the suggestion for Gly-CN as a nucleophile with lower pKa, but as we explained in our response to point 2, we decided to show that ligation can also proceed with the more stable amino acids, even though the rates are significantly slower. From a prebiotic perspective, we believe these results are very relevant and interesting. We have included a remark in the discussion about the likely higher reactivity of Gly-CN, and we have reworded our sentence in line 41 to make our statement more accurate regarding the experiments in ref. 7. We would like to note that in ref. 7. the authors used 3 equiv. of ferricyanide, while here we only used 1 equiv.

The lower number of equivalents of ferricyanide indeed have an effect on the reaction rate. When we repeated the ligation reaction of Ac-SK or Ac-Gly-SH with Gly-CN with one equivalent of ferricyanide, the reaction rate was quite slow and the yield after 1 h was lower than with more equivalents of ferricyanide (see figure below). To our surprise, when we performed the ligation reaction of Gly-CN with Ac-Gly-SH in ferricyanide-pLys(Me)₃ coacervates, it seems that no product was found within 1h, while the ligation of Gly-OH in coacervates did result in product already after 1 h. This contrasts the findings of Powner and co-

workers (Ref. 7), copied below for reference, that Gly-CN outcompetes Gly-OH in a ligation reaction with Ac-Gly-SH because it has a lower pKa. We have included these results in the Supplementary information. It is beyond the current scope of the manuscript to further analyze the reaction rates with aminonitriles, but it suggests that compartmentalization in coacervates can alter the (re)activity of molecules through partitioning and interactions with coacervate components (see also Bauermann et al., JACS 2022), which is one of the main points we wanted to show in our manuscript, and (as far as we know) the first experimental demonstration of such an effect of coacervate-based compartments on chemical reactions.

Figure R4. NMR analysis of the reaction between Ac-SK with Gly-CN (left), and Ac-Gly-SH with Gly-CN (right) in ferricyanide/pLys(Me)₃ coacervates at different ratios of substrates and ferricyanide.

10. Line 48 – 'readily' instead of 'easily'? Also 'guest molecule' – it's not a guest molecule until it is inside the coacervate.

▶ Corrected. Guest molecule is commonly used in the field of protocells and membraneless organelles to indicate the molecule(s) that become taken up by the coacervates.

11. Line 52 – 'overall rate' of...?

▶ We rephrased 'overall rate' to 'overall reaction rates'.

12. Line 71 – 'possible to make coacervates' – use produce or synthesise instead of 'make'?

▶ We rephrased 'possible to make coacervates' to 'possible to produce coacervates'.

13. Line 77 onwards- 'Alternative oxidizing agents, such as ferricyanide ($Fe(CN)_6^{3-}$), could have been essential in the prebiotic activation of building blocks for peptide ligation, 7,9,33,34. The authors have cited ref. 33 on the discussion of ferrocyanide, presumably as a plausible prebiotic substrate. Alongside it they should cite more recent discussions on the formations of ferri- and ferrocyanides, including <https://www.sciencedirect.com/science/article/pii/S0016703719303801> <https://www.science.org/doi/epdf/10.1126/ysciadv.aax3419>

▶ We have added these two citations.

14. Line 81 – '*ferricyanide-catalyzed oxidation reactions*' – are there any *ferricyanide-catalyzed oxidation reactions* in the manuscript? Should we instead say '*ferricyanide-mediated*?' Unless I have missed an *catalytic oxidations* in the manuscript.

▶ The reviewer is right that ferricyanide is reduced during the reaction and not a catalyst. We reworded '*ferricyanide-catalyzed*' to '*ferricyanide-mediated*'.

15. Line 96 – '*client substrate*' – what's that?

▶ Client is commonly used in the field of coacervates and membraneless organelles to indicate molecules that bind to the 'scaffolding' proteins (i.e., the proteins that phase separate and form the mesh of the membraneless organelle). Clients are usually smaller and bind weakly to the scaffold, which means that they are readily exchanged with free client species in the surrounding solution. Here, we used client to emphasize the difference with substrates that become covalently bound. Because we decided to remove the results related to Figure 5, this paragraph in the introduction has now been rewritten and the term client substrate removed.

16. Line 117 – what do we mean by '*salt-free*' solution? I think we mean *NaCl-free* in these cases. The reason why I am a little pedantic about this is that *ferrocyanide* and *ferricyanide* are, by definition, salts.

▶ We used this as a common term in studies on complex coacervates, which are usually composed of large polyelectrolytes. However, the reviewer is right that in this case one of the components, ferri/ferrocyanide is also a common salt. We therefore revised '*salt-free*' to '*NaCl-free*'.

17. Line 119- "*Increasing the length of polylysine to (Lys)₃₀ resulted in the formation of coacervates with trivalent ferricyanide as well (Supplementary Fig. 4c), in agreement with previous studies involving nucleotides.^{19,27}*" What is the reason for ferricyanide now working with longer chains? I think the authors should state this specifically as to how and why this is '*in agreement with previous studies involving nucleotides*'.

▶ We have rephrased this to specifically explain how and why this is in agreement with previous studies on pLys and ADP and ATP.

18. Line 131- '*For example, for samples prepared from 2 mM ferricyanide and pLys (5mM lysine monomers) solutions, we found an internal ferricyanide concentration of ~30 mM, compared to ~0.3 mM in the surrounding aqueous phase. (Supplementary Fig. 5).*' – how long was this concentration maintained within the coacervate? Can the concentrations be modified? For example if you started with 1 mM ferricyanide what are the concentrations within and outside the coacervate, and does addition of ferricyanide to the bulk solution lead to a readjustment of concentrations in the dilute and within coacervate?

▶ Coacervation is a case of liquid-liquid phase separation. The coacervate is therefore in thermodynamic equilibrium with its supernatant, and the concentrations are maintained indefinitely. The concentrations can be modified by changing environmental parameters such as NaCl concentration and temperature, but not by changing the overall concentration of pLys and ferricyanide. Increasing the concentrations of pLys and ferricyanide (at constant ratio) would only increase the total volume of the coacervate phase and reduce the

volume of supernatant (Gibbs phase rule), but the concentrations inside and outside would remain the same. These fundamentals of coacervation have been extensively studied and modeled in previous studies (e.g., Wang et al. *Macromolecules* 2012, <https://doi.org/10.1021/ma301690t>, Priftis et al. *Soft Matter* 2012, <https://doi.org/10.1039/C2SM25604E>, Neitzel et al. *Macromolecules* 2021, <https://doi.org/10.1021/acs.macromol.1c00703>, Bauermann et al. *JACS* 2022, <https://doi.org/10.1021/jacs.2c06265>), and we believe it is beyond the scope of our manuscript to repeat such thermodynamic characterisation. However, because it is relevant for the reactions we describe to know how the concentrations of ferricyanide change in theory, we added a sentence to explain this, with references to the articles cited above.

19. Line 161 – *'From plots of the turbidity we determined the critical salt concentration (CSC), the point at which coacervate droplets completely disappear.'* Can we state specifically what number of the CSC is?

- ▶ We have added the numerical value of the CSC.

20. Line 167- *'we prepared mixtures of $Fe(CN)_6^{4-}$ with pLys and $Fe(CN)_6^{3-}$ with pLys under identical conditions within the highlighted region of Fig. 1g between the two binodal lines.'* – what are the specific 'identical conditions' within the highlighted regions? Can we report concentrations here, or in the SI?

- ▶ We revised this sentence to make it more clear and add a table to state the specific number of CSC and range of salt concentrations for which redox-based switching is possible in Supplementary Table 1.

21. Figure 2. *Why is it that NADH and GSH work at directly opposite pH? Is there a stability issue with the molecules at pH where they don't work? Instead of GSH have the others tried a different thiol e.g N-acetylcysteine or DTT? I am not asking for further experiments here, I am just curious.*

- ▶ This pH effect of the redox reactions was found serendipitously when the reaction between NADH and ferricyanide did not progress as rapidly as we had hoped. When we repeated the reaction at lower pH, we observed much faster formation of coacervates, which puzzled us as well. We attribute the enhanced reactivity of NADH towards oxidation by ferricyanide to the spontaneous saturation of the 5-6 double bond in the pyridinic ring at pH below 7 in NADH (Leduc & Thévenot, 1973, <https://hal.archives-ouvertes.fr/hal-01179282> and Kim & Chaykin, 1969, <https://doi.org/10.1021/bi00846a041>). This saturation coincides with a shift in absorption from 355 nm to 290 nm, and an increase in standard oxidation potential (Leduc & Thévenot, 1974, <https://hal.archives-ouvertes.fr/hal-01179276>). Apparently, electrons are more readily transferred from the hydrated NADH than its unsaturated form. We know that the reaction between NADH and ferricyanide occurs because of the combined increase in turbidity and disappearance of the characteristic yellow color from ferricyanide.

In the case of GSH the ferricyanide reduction occurs more efficiently at pH 7 or above, because the oxidation of the GSH thiol occurs via a deprotonated form. At pH 4, thiol deprotonation is strongly suppressed.

22. Line 265- *'Ferricyanide has been described as a prebiotically abundant oxidizing agent,³³* Please add <https://www.sciencedirect.com/science/article/pii/S0016703719303801> and <https://www.science.org/doi/epdf/10.1126/sciadv.aax3419>

- ▶ We have added these two references.

23. Line 265 '*...and has been used to activate amino thioacids by oxidation to facilitate the formation of an amide bond upon reaction with nucleophilic aminonitriles and amino acids.6– 9,34'* Ref 8. does not use ferricyanide. Ref. 34 does not use amino thioacids. It uses the ferricyanide to oxidise the thiocarbamate that results from the reaction of an amino acid and carbonyl sulfide. This can then cyclise to form an N-carboxyanhydride. The acylating agent is an N-carboxyanhydride.

► Ref. 8 indeed uses ferric chloride instead of ferricyanide as oxidizing agent. In ref. 34, the authors write that during the course of the reaction substantial quantities of H₂S are generated, which can react with the NCA to generate α -amino thioacids that can also participate in the formation of peptides (note 7 in ref. 34). Amino thioacids are thus not directly used, but were likely generated during the reaction. To make our statement accurate, we have reworded the sentence and updated the references.

24. Line 272. '*However, the oxidative aminoacylation of thioacids is usually performed with high reactant and ferricyanide concentrations (sometimes close to 100 mM), which could have been difficult to reach everywhere on Early Earth.*' See comments above about low concentration oxidative peptide ligation in ref. 7. As I said, that does not detract from this work because it's showing the relative rates of ligation within and external to the coacervate. However, the authors should not make the reader think there is a problem that needs to be resolved (low concentration of oxidant for oxidative ligation), rather, the author's manuscript is showing a physicochemical phenomena generated by the coacervate. That is what you should be stressing in your manuscript, which I feel you haven't done forcefully yet but should do in revision.

► We have also revised this statement, in connection to point 9 and we would like to refer to our response to point 9 here. In addition, we have made changes to our manuscript to avoid the suggestion that there is a problem, and instead to emphasize that we show an effect of the coacervate.

25. Generally throughout manuscript – please check carefully where you have used which peptide for coacervation. Eg Supplementary Fig. 10 – did you use Arg₁₀ or pLys?

► In Supplementary Fig. 10 we did the control experiments in both (Arg)₁₀ and pLys based coacervates. We have checked all figures and updated the information on the peptides used where necessary.

Supplementary Information

26. I would recommend that the authors spend time in writing out experimental protocols that reflect what the experimenter did for all experiments discussed within the text and in the figures. I think this should be done as standard at all times. It also makes the experimenter think again whether the procedure they carried out was correctly executed when writing this all up.

Data in figures:

Figure 1. The experiments with NADH, polyu-rU15 and pyranine are not described within the experimental section. The procedures are in the SI right now are so highly generalised. Quantities could be reported so as to reproduce this work. For example, as the reviewer, I cannot assess whether the quantities used in experiments are correct (or not).

Figure 2. These experimental procedures are not described as far as I am aware.

Figure 3. Again, not described.

Figure 4. Not described.

► We have included experimental protocols for all figures in the Supplementary information.

27. Line 339 'In contrast, when Ac-Phe-SH/Phe-SH was added and reacted with the ϵ -NH₂ of pLys, the turbidity transition shifted to a significantly higher salt concentration and the absolute intensity of the plateau at high salt concentration was higher (Fig. 5b).' this reads as Ac-Phe-SH and Phe-SH in the same reaction – I don't think it is. A description of these experiments would resolve ambiguity in the SI.

▶ Please refer to our detailed response to point 4 above, concerning the experiments related to Figure 5. Since we have decided to remove this experiment from the revised manuscript, the ambiguity that the reviewer mentioned is no longer present.

28. Supplementary Table 1 – The selectivity ratios are reported, but what are the actual conversions/yield of ligations?

▶ We revised the Table with the yields of the mixed amino acids experiments in new Table 2, and the time-dependent NMR spectra Supplementary Figure 15-18 are also inserted in the SI.

29. Supplementary Figure 12 – Can we get a number on how much epsilon amine was acylated? Also, the figure shows Ac-Gly-SH, but the text refers to experiments with Ac-Phe-SH.

▶ Please see our response to point 4. The text referring to Supplementary figure 12 contained a mistake – the figure shows the reaction with Ac-Gly-SH. We estimate the ϵ -NH₂ acylation with Ac-Gly-SH to be 10%. As discussed above, we could unfortunately not directly detect the acylation product with Ac-Phe-SH. Since we have decided to remove this experiment from the revised manuscript, this point is no longer present.

Good luck with the revisions, and I look forward to seeing the manuscript again.

▶ We would like to thank the reviewer for her/his detailed and very helpful feedback. We look forward to hearing their response to our revisions.

Reviewer #2 (Remarks to the Author):

Peptide-based coacervates have been widely used as model protocells in the field, however the synthesis of peptides inside of them has not been demonstrated. This is an outstanding problem that the authors attempt to address in this study using molecules that may have been readily available on the primitive Earth. Here they use ferricyanide (Fe^{3+}) and Lysine/arginine coacervates as redox active coacervates. Impressively, they demonstrate that 2mer's made from amino thioacids and peptides/amino acids can be made with fairly high yield by oxidizing the thioacid. Overall this represents a fairly big increase in the field because it presents a chemically feasible way to achieve peptides before invention of the ribosome. There are a number of positive features of this study. For instance, the authors demonstrate that coacervates made from ferri/ferrocyanide PLUS pLys can partition small molecules, which can be used to tune the oxidation state and the coacervation. Moreover this process can be cycled multiple times indicating that oxidation and reduction by alternating additions of GSH and S2O8²⁻ is reversible, as is formation of compartments and homogeneous solution of one phase. Additionally, the authors demonstrate formation of peptide bonds inside of the droplets and that this is selective dependent on the amino acids/amino thioacids involved. Finally, the authors demonstrate novelty in that when hydrophobic amino acids are incorporated the properties of the droplets change.

Major Comments:

1. While this article is about peptides, it is important to think about compatibility with RNA. The authors should test if their coacervate droplets are friendly to RNA or if they degrade the RNA. This will increase the impact and scope of the work if successful.

▶ We thank the reviewer for this suggestion. These coacervate droplets behave very similar to previously reported ATP-PLys coacervates towards RNA and ssDNA. Both oligonucleotides are taken up by the droplets and they are not degraded in the typical timespan of our experiments (24 h). In Figure 1f and S4g, we showed microscope images of coacervates with RNA (poly-U₁₅) and ssDNA (poly-A₁₅), in which it can be seen that both are taken up by the coacervates. The RNA and DNA are not degraded by the droplets, as we see stable fluorescence intensity over time. If the RNA would be degraded, the partitioning would decrease and thus the fluorescence intensity inside the coacervates would decrease over time to background levels.

2. The authors make claims about the liquidity of the droplets in figure 1, in Supplementary figure 8, and in Figure 5. Their observations about wetting of surfaces and coalescence of droplets are good, but FRAP experiments could be done to further demonstrate whether droplets are liquid or gel-like.

▶ We emphasize that complete fusion and relaxation to a spherical shape is only expected for pure liquid droplets, and not for gel-like droplets. Nevertheless, we included FRAP measurements of fluorescently labeled pLys in ferricyanide-based coacervates to show they recover as expected of liquid droplets. We also measured FRAP of the coacervates after reaction with Ac-Phe-SH (Fig. 5), to show that their recovery is significantly impeded because they have become gel-like (results shown below). However, as discussed above, in our response to point 4 of reviewer 1, we have decided to remove the experiments related to Figure 5. While the reaction and measurements of turbidity and FRAP are reproducible, we were not able to directly measure the acylation product of the reaction between pLys and (Ac-)Phe-SH.

Figure R5. FRAP analysis of ferricyanide/pLys30 coacervates and ferricyanide/pLys30 coacervates after incubation with Ac-Phe-SH. Panel on the right shows the distribution of recovery times in seconds.

3. Some discussion of what droplets could do which things would be appreciated. In figure 3 they focus on arg droplets, but for all the rest of the paper it is pLys and its derivatives. Did the experiments not work for the arg droplets?

► Typically, the polycation does not take part in the reactions presented in our manuscript and all types of polycation could be used, although we do observe a faster ligation and higher yield for shorter pLys in Fig.

4. We used pArg in Figure 3, because we noticed that the fibers inside the droplets are better visible in pArg-based droplets. We also performed this experiment with fiber assembly in pLys-based droplets, as shown in Supplementary Figure 8. The reason we did not use pArg-containing droplets for the ligation reactions (Figure 3) is that we did not have fully methylated pArg available.

4. The statement on line 202 says that after 8 cycles the droplets can't be reformed, could this also be because of dilution of the coacervate components and not just of product build up?

► The reviewer is right that dilution could in theory contribute to the fact that the coacervates cannot be formed again after 8 cycles, and we have added this to the text. However, the effect is only very minor: the initial sample volume $V_t = 200 \mu\text{l}$, and we added $1 \mu\text{l S}_2\text{O}_8^{2-} + 2 \mu\text{l GSH}$ for each cycle. Hence, after 8 cycles the volume reached $V_t = 222 \mu\text{l}$, which corresponds to a dilution factor of only 1.11 times.

5. Line 216 They make a statement about fluorescence disappearing from the center of the droplet outward, if possible they should quantitate this phenomenon as it is important to their point.

► We agree with the reviewer and quantified the disappearance of fluorescence in Supplementary fig. 11.

6. In the figure 3 legend, they describe a time series, but don't make it clear on the figure whether microscope images come from different times. Please clarify this.

► The caption has been revised, as this was not a time series, but an image of the final stage. Images of droplets from different times have now been added as Supplementary figure 10.

7. In Supplementary Figure 4h they should include another panel with a bright field image of those same droplets. Also why are there strange lines in supplementary figure 4f?

► We included the corresponding brightfield images for the droplets in Supplementary fig. 4. The lines in 4f are an artefact from the (edge of the) spinning disk of the confocal microscope, which arises when the exposure time is set too close to the time interval of the time series, which was inevitable because of the relatively low fluorescence intensity of NADH. We have replaced the image by a different image taken as a snapshot and removed the artefacts from the spinning disk in the other time series by subtracting a blank image recorded at the same speed.

8. In supplementary figure 8c, the droplets appear to be solid aggregates, they should discuss this.

► These are actually flower-like fibers growing at the surface of coacervates. Similar structures have been reported by Mann's lab (e.g., <https://pubs.rsc.org/en/content/articlepdf/2016/sc/c6sc00205f>). The resulting composite is probably solid-like, and we have included a discussion of their state in the manuscript.

9. Line 284, did the researchers do the peptide bond experiment in ferrocyanide-pLys(Me)₃ coacervates? As this would be a direct comparison to their other experiment in ferricyanide-pLys(Me)₃. If this doesn't form a phase, it makes sense, but they should state that.

► In Supplementary figure 13, we show a control experiment of the ligation reaction in ferrocyanide-pLys. This combination does form droplets, but because no ferricyanide (the Fe(III) species) is present to oxidize the aminothioacid, the ligation cannot take place, and we observe no decrease of the thioacid signal in NMR. This sample thus serves as a (negative) control to show that the ligation is catalyzed by ferricyanide.

10. Line 316 they make a statement about kinetic pathway selection when there is Gly and Glu there. Some discussion of how this compares to Glu only would be helpful here.

► We have included a discussion of the control experiments with Glu only. Moreover, in relation to points raised by reviewer 1, we have expanded this discussion with experiments with other mixed amino acids.

a. Also some mention of the N of experiments would be good. If N is 1, suggest completing at least 2 more replicates.

11. In general, they should give the # of experiments in the figure legends for all of the experiments that they did. This is particularly lacking in figures 2, 4, and 5.

► We have included the number of repeat experiments in all Figure legends.

Figure	# of experiments
1g	1
2e	3
3c	2 for (LysMe) ₂₀ and (LysMe) ₃₀ each 4 for LysMe 15-30K
3d	3
4b	3
S8	1

S9	2
S12-13	1
S15-18	1 (But different time points were different samples)
S19	2
S20	1

12. Figure 5C they state these droplets are gels, consider doing FRAP to demonstrate this. Additionally, some type of statistics should be given to know that this image is representative of the entire field.

▶ As mentioned in our response to point 2, we have added FRAP measurements to prove that the droplets are indeed gel-like. However, as discussed above (response to point 4, reviewer 1, and point 2, reviewer 2), we have decided to remove the experiments related to Figure 5 from the revised manuscript, because the acylation product of pLys and (Ac-)Phe-SH could not be measured directly. Therefore, this point is no longer present in the revised manuscript.

13. Supplementary Figure 6, it would be helpful to have some quantitation of the turbidity. e.g. See Fig 3 in reference 19.

▶ The turbidity in Supplementary fig. 6 was defined as the absorbance at 520 nm and therefore has no unit. It can be converted to % as described in Nakashima et al., Meth. Enzym. 2020. We have converted the turbidity now to %.

Minor comments

1. They should be consistent about use of the short form of ferri/ferrocyanide. Seems random now.

▶ We agree about the importance of consistency and have used $\text{Fe}(\text{CN})_6^{3-}$ and $\text{Fe}(\text{CN})_6^{4-}$ as short forms of ferricyanide and ferrocyanide, respectively. Only in full-text sentences where it made sense for readability to keep the long form of ferricyanide and ferrocyanide, we have kept these.

2. In figure 3 or in the text some discussion of the time required to form fibrils would be good.

▶ We have included this in the discussion, the time required to form fibrils is about 5 minutes.

3. Line 282 be specific about which coacervates. There are many described in the paper and the next line there is talk of a different coacervate.

▶ We have added specifics of the coacervates.

4. Line 290 it would give the readers more context to either give the mass of the shorter Lys derivatives or give the dispersion of sizes of the pLys because their point is that the short ones do the reaction better.

▶ We have added the mass of the shorter pLys to make comparison with the long 15-30k pLys more straightforward.

5. In Figure 4, they show formation of a 2mer inside of the coacervates. Is their evidence that longer peptides can be made given addition of more substrates?

▶ With the substrates we used in for the reaction in Fig. 4, only dipeptides can be formed: the thioacid has an acetyl-capped N-terminal end, while the amino acid is not activated (e.g., thioacid or thiocarbamate). Further reaction of the dipeptides would require reactivation by thiolysis, as described in the reaction cycle in ref. 7. Addition of more substrate would not be sufficient to generate longer peptides.

6. State what GSH means in Figure 5 figure legend.

▶ We have written down the abbreviation in the caption of Fig. 5.

7. Supplementary Information should have a table of contents at the beginning to make it clear what is where in the SI.

▶ We have added a table of contents to the Supporting information.

Reviewer #3 (Remarks to the Author):

In this manuscript, Jiahua Wang et al. utilize coacervate droplets made from redox-active ferricyanide (Fe^{3+}) or ferrocyanide (Fe^{2+}) and polylysines or oligoarginine as protocell models. With the addition of a reducing or oxidizing agent, they can switch between the phase-separated and the mixed state for multiple cycles. Interestingly, both assembly and disassembly can be performed at either low or neutral pH, dependent on the reducing agent. They show that they can use the oxidizing properties of ferricyanide to trigger fiber assembly inside the droplets via oxidation of a benzoyl thiol. Finally, they use the propensity of ferricyanide to accelerate oxidative aminoacylations of thioacids to dimerize compartmentalized amino acids. They show that some amino acid dimers are selectively formed in favor of others and they claim that this is due to different binding affinities to the droplet's building blocks. In some cases, the droplet building blocks themselves are modified so that the resulting compartment becomes more salt-resistant which the authors propose as an advantage in terms of protocell fitness.

In general, the results shown here are promising and could be of great value in the origin of life field. Most reported compartmentalized reactions in coacervate droplets focus on enzymes or ribozymes but synthesis of peptide building blocks is indeed scarce. Also, in terms of redox-active coacervates as protocell models little is reported so far. I do, however, have several major and minor concerns:

1. Mechanism of selective amino acid dimerization. The authors argue that glycine is the most reactive amino acid since it has the lowest affinity to polylysine, one of the two main droplet building blocks. If so, it seems counter-intuitive to me that glutamic acid with its negative charges is more reactive than phenylalanine, which in turn should have weaker interactions with the polycation than glutamate. Also, using only glutamic acid, the dimer Gly-Glu is formed with comparable rates than Gly-Gly (in case of 100% Gly), indicating that the polylysine interaction is not decisive here. Therefore, the proposed mechanism seems to be rather vague and not fully explain what is going on. I wonder whether other parameters are much more important, eg. sterically hinderance. One could test this maybe with other bulky amino acids? Ideally one could use basic amino acids such as lysine or arginine to have a control for the glutamic acid. But I assume that the side chain would react as well. so methylated lysine or arginine would be needed? One could also try to modulate the affinity towards glutamic acid by lowering the pH or using a protected derivative.

► We thank the reviewer for their comments. The reviewer is right that the reason(s) for the selectivity are complex: the amino acids partition into the coacervates to a different extent (thus changing the effective local concentration), they interact with the peptides inside the coacervate to a different extent, and there may be different steric hindrance inside the dense coacervate. We have performed additional experiments with a range of amino acids, also following suggestions by reviewer 1, and present the results in Supplementary fig. 21-32 and Table 2. We have also included relevant parameters, such as the amino acid pKa and effective radius, and their partitioning inside the coacervates in Supplementary fig. 20 and Table

2. As can be seen, the reason for the observed selectivity is not simply affinity, or steric hindrance, but most likely a combination of factors. We have therefore rephrased the discussion linked to these experiments and to Fig. 4.

2. *Enhancement of ligation in droplets with shorter peptides. The authors attribute this effect to the lower multivalency of the polycation, which in turn would result in weaker complexation of the ferricyanide. What about other effects such as diffusivity of the molecules inside of the droplet which is directly linked to the length of the polymers. Techniques like FRAP would help here.*

▶ We agree with the reviewer that the diffusivity of the reacting molecules and oxidizing agent (amino (thio)acids, ferricyanide) inside the droplet may be of relevance for the enhancement of ligation with shorter peptides. However, determining the diffusivity of these molecules (aminothioacid, amino acid, ferricyanide) would require labeling them with fluorophores, which would completely change their properties and possibly impede their reactivity. The diffusivity of the peptide itself can be measured by FRAP, and we have included FRAP data of the peptide in coacervates in Supplementary figure 3. However, mobility of the peptide does not necessarily reflect the diffusivity of small molecules inside the coacervates.

3. *Protocell fitness. The authors argue that the salt-resistance from side-rctns that modify the droplet's main building blocks may be advantageous in terms of protocell fitness. I would agree with that until a certain point. If the metabolism inside a protocell gets out of control, one will not get a fitter compartment but just an agglomerate that has lost its function. One of the reasons why coacervates are regarded as relevant protocell models is because they have considerable water inside, so the molecules inside the droplet can move rapidly. Again, FRAP would be a good first indication of how fluid the droplets still are. Also, it would be nice to see the new CSC of the "fitter" droplets since it is not shown in Fig 5b.*

▶ We agree with the reviewer that our argument is only true up to a certain point. We have performed FRAP on the gel-like droplets in Fig. 5 and found that the peptides do not recover (results shown above in our response to point 2). The CSC of the side-chain modified coacervates were shown in the inset of Fig. 5b. However, as discussed above (response to point 4, reviewer 1, and point 2, reviewer 2), we have decided to remove the experiments related to Figure 5 from the revised manuscript, because the acylation product of pLys and (Ac-)Phe-SH could not be measured directly. Therefore, this point is no longer present in the revised manuscript.

4. *Reversibility. In Figure 2, the authors explain that one can switch between assembly and disassembly via reduction of ferricyanide and oxidation of ferrocyanide. However, in Figure 5b and Supplementary Figure 7, the reduction of ferricyanide to ferrocyanide dissolves the droplets. This is not intuitive because of the increase in charge density and valency. Something seems off.*

▶ The experiment was unfortunately not explained clearly enough. In Fig. 5b, we added Phe-SH and Ac-Phe-SH to the ferricyanide/pLys coacervates, and observed a change in color of the ferricyanide/pLys dispersion from yellow to white turbid, and the overall turbidity increased, which means that more stable ferrocyanide coacervates were formed. As a control, we added GSH (a reducing agent) to the ferricyanide/pLys coacervates to form ferrocyanide without modifying the pLys. After that, we performed a salt titration in order to dissolve the reduced ferrocyanide coacervates and determine their CSC. We have rephrased the description of this experiment to clarify what caused the dissolution of the coacervates.

Supplementary figure 7 only shows the fluorescence channel. In this case, the droplets do not disappear, but the fluorescence from NADPH does, as it is oxidized to NADP⁺, which is not fluorescent at the selected wavelength. We have clarified this in the figure caption.

Minor comments:

1. *Since there are no microscopy images in Figure 4. Are the droplets affected in any way by the reaction inside?*

▶ We have added microscope images of the droplets during the reaction, they are not affected during the reaction (Supplementary fig. 33). The droplets grow slightly as a result of coalescence, which happens to any coacervate dispersion upon incubation.

2. *In S7, the fluorescent "holes" appear due to an oxidation rctn. In Fig 1f and S4, however, these holes are also present. Why?*

▶ In fig 1f, and S4 the relatively large dyes were not able to penetrate to the core of the droplets. Smaller dye molecules, such as NADH, NADPH and pyranine (Fig. 1d, e) can penetrate to the core of the droplets. Similar observations have been made previously by Matsuo and Kurihara (<https://www.nature.com/articles/s41467-021-25530-6>).

3. *In S9&10, spectra of later time points would be good, similar as in Fig 4b. Also, the peaks in S9&10 are slightly shifted. Is that due to different deuterated solvents?*

▶ We have added spectra of the sample after 3 days in Supplementary fig. 12 to show that no significant reaction takes place even after longer time. The peaks in Fig. S9 and 10 are slightly shifted because of the presence of paramagnetic ferricyanide in S9.

4. *Figure 5b: caption misses what type of coacervates*

▶ This figure has been removed (see discussion above in our response to point 4, reviewer 1, and point 2, reviewer 2).

5. *In S12, the ligation of glycine to polylysine is shown. However, in the text and in Figure 5, the ligation of phenylalanine is discussed. Also, should not there be more orange proton peaks?*

▶ Please refer to our response to 29, reviewer 1. With respect to the number of proton peaks: the ligation of Ac-Gly-SH only yields two detectable orange proton peaks (the acetyl group and alpha-CH₂, respectively).

We thank all reviewer for their valuable feedback and look forward to their comments on our revised manuscript.

REVIEWERS' COMMENTS

Reviewer #1 (Remarks to the Author):

The authors have comprehensively answered all the points I raised in my review, and I have no further questions. I also assessed their answers to the other reviewers, and they appear satisfactory. My thanks to the authors who invested the time and effort to carefully resolve ambiguities from the first round. I recommend publication in Nature Communications.

Reviewer #2 (Remarks to the Author):

The authors have address many of the concerns, but there are still some issues that need addressing.

- 1.) Methods do not mention the timescale of the reactions.
- 2.) Its nice to have FRAP experiments. However, as far as I know a $t_{1/2}$ of 5 sec is still quite liquid rather than gel-like.
- 3.) Are the authors confident that all types of polycation could be used as they only tested 2. The callout they use here is not correct as far as I can tell, sup fig 8 is about partitioning of amino acids. I
- 4.) Good.
- 5.) I like that they did this. However what is the fit on the graph? Is there a standard error of the fluorescence? How many droplets were measured at each time? Information is needed in the legend and statistics in tables.
- 6.) As far as I can tell there is no images in figure 3 in the main text anymore and the sup fig 10 callout also has none. They did remove the reference to a time series.
- 7.) This is fine but the authors need to call out which main text figure these are the corresponding bright fields for in the legend of this figure.
- 8.) I was disappointed that the authors didn't put the discussion of this in their rebuttal, however they did cite an appropriate paper.
- 9.) Good
- 10.) Addressed
- 11.) Many errors in the callouts and it seems the experiments are often single attempts and not in triplicate. Also, number of droplets observed should be reported.

12.) Good.

13.) Good.

Overall, the SI needs more organization. All callouts should be checked.

► *Reviewer's comments are greyed out and italicized; authors' responses are in black font right below each group of comments, as well as highlighted in the manuscript in blue.*

Reviewer #1 (Remarks to the Author):

The authors have comprehensively answered all the points I raised in my review, and I have no further questions. I also assessed their answers to the other reviewers, and they appear satisfactory. My thanks to the authors who invested the time and effort to carefully resolve ambiguities from the first round. I recommend publication in Nature Communications.

► We would like to thank the reviewer for their assessment of our manuscript.

Reviewer #2 (Remarks to the Author):

The authors have address many of the concerns, but there are still some issues that need addressing.

1. Methods do not mention the timescale of the reactions.

► We have added the timescale of the reactions in our methods.

2. It's nice to have FRAP experiments. However, as far as I know a $t_{1/2}$ of 5 sec is still quite liquid rather than gel-like.

► We agree with this assessment, but note that this point is no longer relevant, as we have removed the old Figure 5 and corresponding supplementary information from the manuscript, because we could not directly show the formation of a peptide bond between the epsilon-amine and Ac-Phe-SH. To comment on the reviewer's point, we assume this is a consequence of the fact that the degree of modification of the pLys scaffold is very low. Gel-like may therefore not have been the best term to use. However, as we have also removed the FRAP results of the coacervates after incubation with Ac-Phe-SH from the Supplementary Information, and we do not mention gel-like coacervates in the manuscript anymore. The FRAP data in Supplementary Figure 3 therefore only contains the original ferricyanide-based coacervates, which have are liquid droplets with a short $t_{1/2}$ of 1 sec.

3. Are the authors confident that all types of polycation could be used as they only tested 2. The callout they use here is not correct as far as I can tell, sup fig 8 is about partitioning of amino acids.

► We apologize for the mistakes in the callout. We also performed this experiment with fiber assembly in (Arg)₁₀, (Lys(Me)₃)₂₀, (Lys(Me)₃)₃₀ and pLys-based droplets, as shown in Supplementary Figure 34.

4. Good.

► No action needed.

5. I like that they did this. However, what is the fit on the graph? Is there a standard error of the fluorescence? How many droplets were measured at each time? Information is needed in the legend and statistics in tables.

► As suggested, we have updated our legend to include the information regarding our data analysis conditions. The resulting experimental data were fitted to an exponential growth function of the form $R = 1 - e^{-(t-t_0)/\tau}$, where τ is the characteristic timescale of diffusion. We averaged the radius increase of 5 droplets at each time point.

6. As far as I can tell there is no images in figure 3 in the main text anymore and the sup fig 10 callout also has none. They did remove the reference to a time series.

► We thank the reviewer pointing this out. Formerly, what labeled as Figure 3 has been renumbered as Figure 4, and we have included images captured at different time points as Supplementary figure 33. We have corrected this mistake in the callout.

7. This is fine, but the authors need to call out which main text figure these are the corresponding bright fields for in the legend of this figure.

► As suggested, we have updated the legend, pointing out each corresponding figures in the main article.

8. I was disappointed that the authors didn't put the discussion of this in their rebuttal, however they did cite an appropriate paper.

► We acknowledge that we did not include a discussion of their state, as was suggested by our reply. We have now added a sentence to our manuscript to give some more context to these structures.

9. Good

► No action needed.

10. Addressed

► No action needed.

11. Many errors in the callouts and it seems the experiments are often single attempts and not in triplicate. Also, number of droplets observed should be reported.

► We have corrected the errors in the legend. For amide bond formation within the coacervates, each time point represents an independent sample, and we have now clarified this in the supplementary methods. At every time point during the reaction, we quenched one reacting sample, and added enough salt to ensure complete dissolution of the coacervates and coacervation adhesive on the sample tube, facilitating optimal conditions for the NMR measurement. Where relevant, we have also indicated the number of droplets analyzed.

12. Good

► No action needed.

13. Good

► No action needed.